# Preclinical assessment of the efficacy and specificity of GD2-B7H3 SynNotch CAR-T in metastatic neuroblastoma

Babak Moghimi[1,2], Sakunthala Muthugounder[1], Samy Jambon [1], Rachelle Tibbetts[1], Long Hung[1], Hamid Bassiri [3], Michael D. Hogarty [3], David M. Barrett[3], Hiroyuki Shimada[1,2] & Shahab Asgharzadeh[1,2✉]

The ability to utilize preclinical models to predict the clinical toxicity of chimeric antigen receptor (CAR) T cells in solid tumors is tenuous, thereby necessitating the development and evaluation of gated systems. Here we found that murine GD2 CAR-T cells, specific for the tumor-associated antigen GD2, induce fatal neurotoxicity in a costimulatory domain-dependent manner. Meanwhile, human B7H3 CAR-T cells exhibit efficacy in preclinical models of neuroblastoma. Seeking a better CAR, we generated a SynNotch gated CAR-T, GD2-B7H3, recognizing GD2 as the gate and B7H3 as the target. GD2-B7H3 CAR-T cells control the growth of neuroblastoma in vitro and in metastatic xenograft mouse models, with high specificity and efficacy. These improvements come partly from the better metabolic fitness of GD2-B7H3 CAR-T cells, as evidenced by their naïve T-like post-cytotoxicity oxidative metabolism and lower exhaustion profile.

[1] Children's Hospital Los Angeles, Children's Center for Cancer and Blood Diseases, Division of Hematology, Oncology and Blood & Marrow Transplantation, and The Saban Research Institute, Los Angeles, CA, USA. [2] Keck School of Medicine, University of Southern California, Los Angeles, CA, USA. [3] Children's Hospital of Philadelphia and University of Pennsylvania School of Medicine, Philadelphia, PA, USA. ✉email: sasgharzadeh@chla.usc.edu

While the results of CAR-T-cell therapy in B-cell malignancies are highly encouraging, the treatment of pediatric solid tumors with a similar approach has shown limited efficacy[1,2]. The barriers to improving the success of CAR-T cells for solid tumors are multifactorial. Among these challenges is the identification of tumor-associated antigens (TAA) with minimal off-tumor side effects. Unlike CD19 CAR-T cells, where elimination of normal B cells can be medically supported, low expression of TAA in other normal tissues can instigate devastating effects in the presence of potent T cells that are not easily remediated[3]. To date, CAR-T-cell clinical trials for adult solid tumors have been associated with severe toxicities or had little efficacy[4]. The toxicities associated with these therapies have led to the design of terminating switches that, when activated, can efficiently eliminate the CAR-T cells but at the cost of reduced efficacy[5–8]. A novel approach using synthetic Notch (SynNotch) design was recently described as a gating strategy where expression of a CAR for one TAA is dependent on initiation of a transactivating signal by another TAA[9,10]. This approach fuses a single-chain variable fragment (scFv) directed against a TAA to a SynNotch receptor, thus creating the gate. Upon binding of the non-gated CAR to the first TAA, the SynNotch site is cleaved, releasing an intracellular nucleolus-bound transcriptional activator that induces gated expression of a CAR against the second TAA. Expression of the gated CAR is initiated by an upstream activation sequence (UAS) that has an exclusive binding site for the transcriptional activator from SynNotch. Thus, the expression of the second CAR is dependent on its gate, and maximal CAR-T cytotoxic activity is dependent on the presence of the second TAA. After the disengagement from the gate antigen, the expression of the gated CAR will decay[10]. We hypothesize that a SynNotch-gated strategy could generate specific and efficacious CAR-T cells against neuroblastoma (NBL), a common solid tumor of childhood. Furthermore, we hypothesize that the intermittent gate-dependent expression of the second CAR in this design will lead to less tonic signaling resulting in less T-cell exhaustion and improved metabolic fitness.

To test our hypothesis, we built a gated CAR-T-cell targeting NBL and evaluated its safety, specificity, and efficacy against NBL cell lines and murine models. NBL, a neuroendocrine tumor, is the second most common solid tumor of childhood and has extremely poor survival in children identified with high-risk features. Antibodies directed against disialoganglioside (GD2), a TAA found on NBLs, melanomas, and sarcomas, are routinely used in upfront and relapse NBL therapies and have improved outcomes for children with high-risk disease[11]. GD2 is also expressed at low levels on neurons and peripheral nerve fibers, with excruciating pain being the most common toxicity associated with anti-GD2 antibody infusion, routinely necessitating concomitant infusion of opioids[12,13]. Rare motor neuropathies have also been described in adults[12–14] (Dinutuximab injection, for intravenous use: US prescribing information, 2015, http://www.fda.gov/). Interestingly, clinical trials with CAR-T cells directed against GD2 have not shown evidence of pain or neurotoxicity; however, their efficacy has also been negligible[15,16]. In a xenogeneic murine model of NBL, CAR-T cells constructed with a mutated high-affinity variant of anti-GD2 (E101K) scFv showed improved in vitro and in vivo efficacy but at the cost of significant neurotoxicity and death in treated immunocompromised mice[17,18]. Given the high prevalence of GD2 expression in NBL[19], identification of a high-affinity anti-GD2 (E101K) scFv with the promise of improved efficacy, and the need to avoid potential neurotoxicity, we chose GD2 as the gating TAA in constructing our SynNotch receptor.

B7H3 (CD276) was chosen as the TAA for the CAR construct in our SynNotch system. B7H3 is an immune checkpoint molecule expressed at high levels on several adult and pediatric solid tumors, including sarcomas, brain tumors, and NBLs[20–22]. In addition to the expression on tumor cells and tumor vasculature, B7H3 is expressed at variable levels on some normal tissues, including sinusoidal endothelial cells of the liver, prostate, adrenal gland, and activated monocytes, depending in part on the antibody used for immunohistochemical or flow cytometry analysis[23,24]. Recently, B7H3 CAR-T cells showed success in preclinical models of pediatric sarcoma, medulloblastoma, NBL, and adult tumors[20,25]. Preclinical studies of enoblituzumab, an Fc-enhanced humanized anti-B7H3 antibody currently in phase 1 trials, have shown a delay in the growth of different B7H3-expressing primary tumors[26]. Radiolabeled 8H9, another anti-B7H3 antibody, has also demonstrated significant efficacy in the treatment of brain tumors and of NBLs metastasized to the brain[27].

Here, we show that fatal neurotoxicity in mice treated with murine GD2 CAR-T cells is dependent on the costimulatory domain used in their construct. We show that safer GD2 and B7H3 CAR-T cells using the SynNotch strategy display a remarkable degree of therapeutic discrimination in vitro and in vivo—sparing single antigen "bystander" cells while eradicating GD2$^+$B7H3$^+$ NBL cells. GD2-B7H3 CAR-T cells maintain high metabolic fitness comparable to resting T cells, are more resistant to exhaustion, and have better in vivo efficacy post exhaustion compared to conventional B7H3 CAR-T cells. This study demonstrates the safety and functional advantages of gated CAR-T cells in solid tumors where toxicity from conventional CAR-T cells is a major concern.

## Results

**Fatal neurotoxicity in mice treated with GD2-28z murine CAR-T cells.** We initially set out to develop murine GD2 (mGD2) CAR-T cells with wild-type GD2 scFv containing either murine CD28 (mGD2-28z) or 4-1BB (mGD2-BBz) costimulatory domains (Supplementary Fig. 1a) to evaluate and compare their efficacy in immunocompetent and immunodeficient NBL models. In vitro, both mGD2 CAR-T cells showed significant proliferation, cytokine production, and specific tumor lysis in the presence of murine NBL cell lines (Supplementary Fig. 2a–f). To our surprise, in vivo treatment with mGD2-28z but not mGD2-BBz CAR-T cells following a lymphodepletion protocol resulted in significant neurotoxicity and death in nearly all immunocompetent and immunodeficient tumor-bearing mice (Figs. 1a and 2a). CAR-T cells were only detected in the brain of mGD2-28z CAR treated animals (Figs. 1b, c, 2b, and Supplementary Fig. 3a, b). Animals treated with mGD2-28z CAR showed signs of reduced tumor burden before succumbing to neurotoxicity, while those treated with mGD2-BBz CAR had no evidence of neurotoxicity and minimal anti-tumor efficacy, with only 1 out of 11 mice showing a decrease in tumor signal (Fig. 1a and 2a). Similar to the result from mGD2-28z CARs, fatal neurotoxicity has been previously demonstrated in an immunodeficient animal model of NBL treated with human CAR-T cells harboring a mutated high-affinity scFv to GD2 (anti-GD2$^{E101K}$) but not its wild-type counterpart[18]. Our data demonstrate that the choice of a costimulatory domain in an otherwise identical CAR-T-cell construct can lead to neurotoxicity. The unpredictable toxicity seen with murine GD2 CARs compelled us to build gated systems directed against GD2 and B7H3 and evaluate their specificity, efficacy, and toxicity in NBL models.

**B7H3 CAR-T cells show effective anti-tumor activity in several NBL models.** B7H3 is highly expressed in many pediatric solid tumors, with the majority of NBL having some positivity for

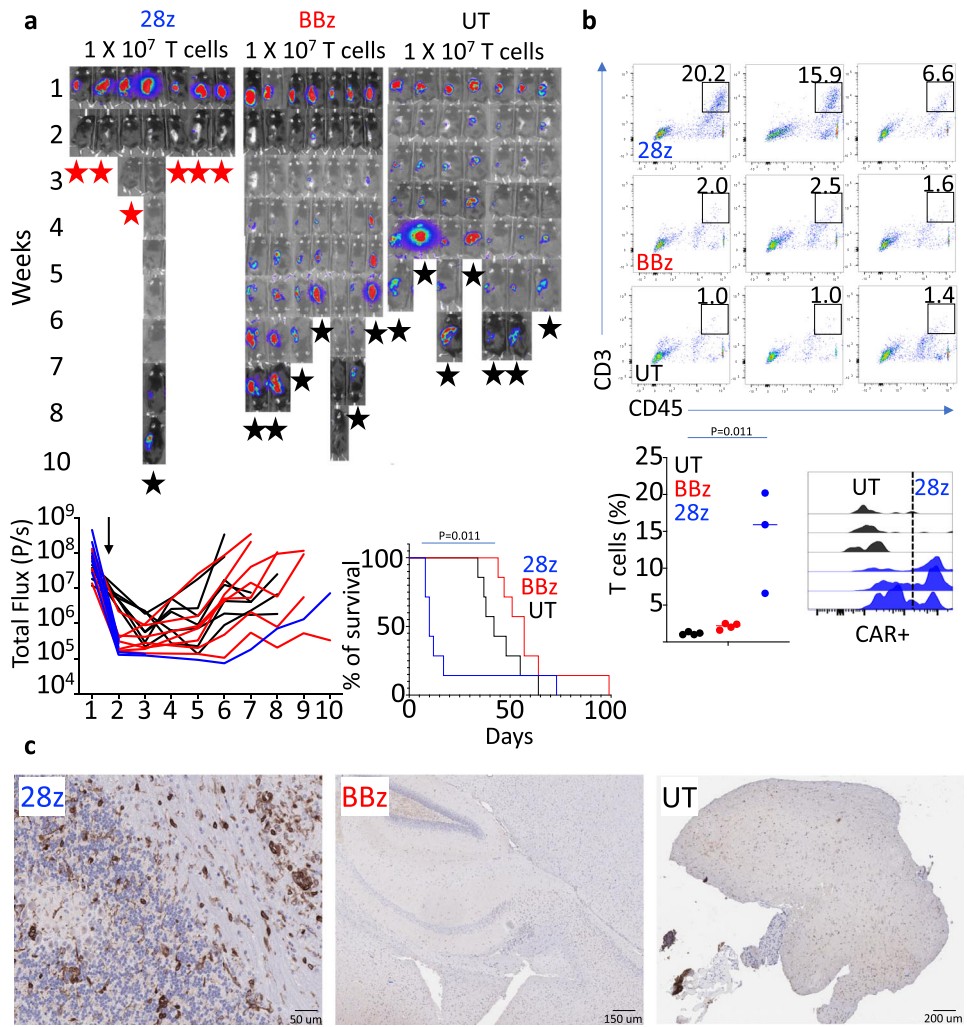

**Fig. 1 GD2-28z murine CAR-T cells cause fatal neurotoxicity in immunocompetent mice. a** Bioluminescence images from a survival study of NB9464D[GD2+Luc+] tumor-bearing mice treated with a 5-day course of chemotherapy followed 72 h later with GD2-28z (28z), GD2-BBz (BBz), or untransduced (UT) murine T cells. Animals were injected intravenously with GD2-28z, GD2-BBz, or UT murine T cells (1×10[7] cells/mice) at 72 h post completion of chemotherapy. Six out of seven animals treated with murine GD2-28z CAR-T cells experienced significant toxicity (seizure, hunched, and immobile) 5–8 days after CAR-T infusion and were either immediately sacked or were found dead. The remaining animals died of tumor growth at various timepoints with an average of 6 weeks post start of chemotherapy, except for one animal in the GD2-BBz group. (Red star—death from neurotoxicity, black star—death from tumor). Lower left: Individual bioluminescence intensity of NB9464D[GD2+Luc+] tumor-bearing immunocompetent mice starting from the week before the administration of chemotherapy and murine GD2-28z, GD2-BBz CAR-T, or UT cells. All GD2-28z animals, except for one, were found dead or euthanized for evidence of severe neurotoxicity. The black arrow points to the time of CAR-T or UT T-cell injection. Lower right: Kaplan–Meier survival graph for the survival study of animals treated with murine GD2-28z, GD2-BBz, or UT T cells. **b**, upper and lower left: Flow cytometry dot plots and T-cell frequency from single-cell dissociated brain tissue from groups of mice treated similar to groups used in the survival study of Fig. 1a, but who were euthanized upon the onset of neurological symptoms in the GD2-28Z-treated group. Lower right: Histogram representing CAR-T cells identified among CNS-infiltrating CD3[+] T cells using an anti-Fab antibody. **c** Immunohistochemical analysis of murine CD3 (brown) in brain tissue of CAR-T-cell-treated animals. $n=7$ mice (28z, BBz, or UT) (**a**), $n=4$ mice (28z, BBz or UT) (**b**). Gehan–Breslow–Wilcoxon test (**a**). Two-tailed $t$ test (**b**). Experiment (**b**) performed independently from (**a**). The data shown are representative of three individual mice from each group, remaining images are included in the Supplementary Information (**c**). Source data are provided as a Source Data file.

B7H3[20]. We evaluated cell surface antigen density of B7H3 and GD2 in human NBL cell lines (LAN6, CHLA51, SMS-SAN, LAN5, SK-N-BE(2), CHLA255). We found high expression of B7H3 and GD2 across both MYCN amplified and non-amplified cell lines except for one cell line (LAN6) that expressed B7H3 but lacked expression of GD2 (Fig. 3a). CAR-T cells generated using anti-B7H3 scFv fused to 4-1BB and CD3z (Supplementary Fig. 1b) showed significant in vitro proliferation, cytokine production, and specific tumor lysis in the presence of B7H3[+] but not B7H3[-] cells (Fig. 3b–f and Supplementary Fig. 4a–d). Also, in vitro, B7H3 CAR-T cells but not untransduced T cells (UT)

demonstrated B7H3-specific CD107a degranulation and intracellular expression of cytokines (IL2, IFNγ, and TNF) when cocultured with NBL cells for 24 h (Fig. 3b, c and Supplementary Fig. 4a). Complete eradication of NBL cells by day 5 was associated with significant B7H3 CAR-T-cell expansion, as demonstrated by an absolute fold increase in T-cell count using carboxyfluorescein succinimidyl ester (CFSE) assay (Fig. 3d). B7H3 CAR-T cells also showed significant secretion of effector cytokines, including GM-CSF, IFNγ, IL2, MIP1b, and TNF in the presence of NBL cells (Fig. 3e). Time-course cytotoxicity analyses of B7H3 CAR-T cells showed potent cytotoxicity against

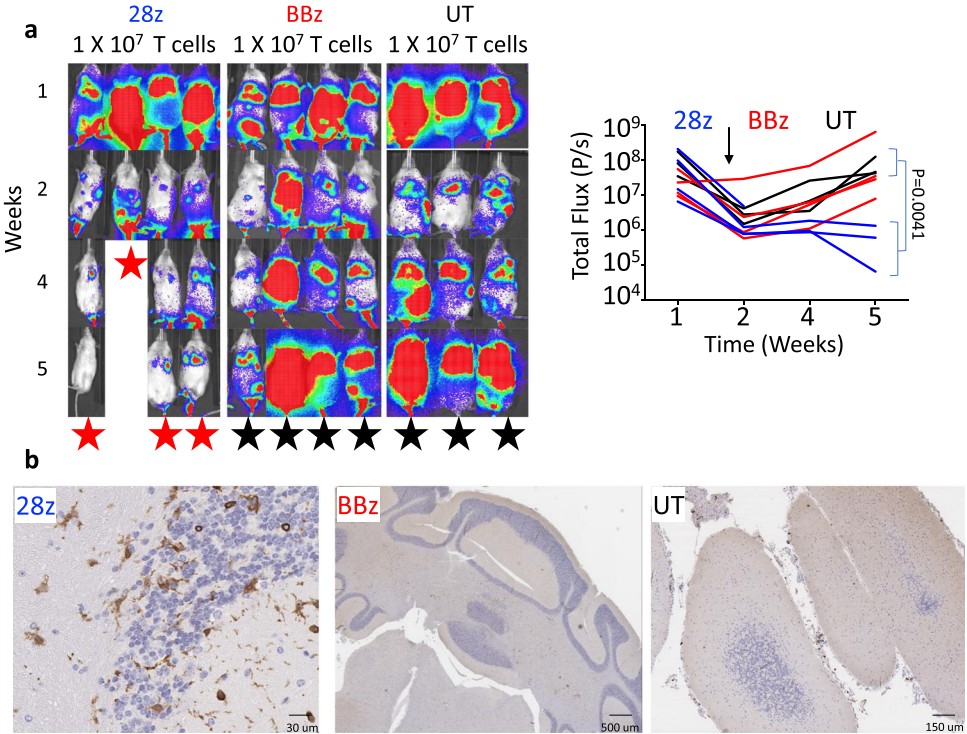

**Fig. 2 GD2-28z murine CAR-T cells cause fatal neurotoxicity in immunodeficient mice. a**, left: Representative bioluminescence images and (right) bioluminescence intensity line plot of the NB9464D$^{GD2+Luc+}$ tumor-bearing NSG mice treated with a 5-day course of chemotherapy followed 72 h later with GD2-28z (28z), GD2-BBz (BBz), or UT murine T cells. The black arrow points to the time of injection of CAR-T or UT T cells. All four animals treated with murine GD2-28z CAR-T cells experienced significant toxicity (seizure, hunched, and immobile) 7–21 days after CAR-T infusion and were either immediately euthanized or were found dead. Animals from other cohorts euthanized for tumor growth at various timepoints by 5 weeks post start of chemotherapy. (red star—death from neurotoxicity, black star—death from the tumor) **b** Immunohistochemical analysis of murine CD3 (brown) in brain tissue of CAR-T-cell-treated NSG mice. The data shown are representative of three individual mice from each group (**b**). n = 4 mice (28z or BBz), n = 3 (UT). Two-tailed *t* test (**a**). Source data are provided as a Source Data file.

CHLA255, LAN5, and SK-N-BE(2) at T-cell effector to target cell (E:T) ratios ranging from 2:1 to 20:1 with no cytotoxicity seen with UTs (Fig. 3f) accompanied by CD107a degranulation in a direct co-culture system (Supplementary Fig. 4a). We then utilized a xenograft model of progressive metastatic NBL by injecting $1 \times 10^6$ luciferase$^+$ CHLA255 cells intravenously into NSG mice. Serial bioluminescent imaging (BLI) following injection demonstrated tumor engraftment in the liver, bones, and brain and subsequent fatality within five weeks post-injection. Tumor-bearing mice injected with $1 \times 10^7$ B7H3 CAR-T cells at 14 days post-tumor inoculation showed complete and durable eradication of tumor, leading to 100% overall survival over the 6-month observation period, while mice that received UT cells or no cells died within 1 month of tumor inoculation (Fig. 3g). Similar in vivo efficacy of B7H3 CAR-T cells was observed in a second metastatic murine model with an *MYCN* amplified NBL cell line CHLA136 (Supplementary Fig. 4e). Immunohistochemical evaluation of liver tissues of mice with the high-burden disease (day 28 post-tumor inoculation) euthanized 7 days post B7H3 CAR-T-cell infusion revealed impressive T-cell infiltration and tumor reduction compared to mice treated with UT cells (Fig. 3h). In summary, our data suggest that conventional B7H3 CAR-T cells are highly effective against NBL and build upon previous observations demonstrating efficacy in vivo against *MYCN* amplified subgroup of NBL.

**Construction of highly specific and effective gated CAR-T cells against NBL.** To evaluate the GD2-SynNotch receptor as a gate for B7H3 CAR expression (Fig. 4a, e), we first engineered the GD2-SynNotch receptor together with a blue fluorescent protein (BFP) construct with a Gal4-VP64 UAS element (Fig. 4b and construct Supplementary Fig. 1b). We incorporated high-affinity E101K-mutated GD2 scFv to increase the potency of the gate sensor (GD2$^{E101K}$ SynNotch). Human primary T cells transduced to express the GD2$^{E101K}$ SynNotch-gated BFP (GD2-BFP) showed activation only in the direct presence of GD2$^+$ NBL cells, with plateauing of BFP expression occurring within 48 h (Fig. 4c and Supplementary Fig. 5a). BFP expression was tightly regulated by the GD2$^{E101K}$ SynNotch receptor, and GD2-BFP T cells exposed to GD2$^-$ NBL cells (LAN6) did not show significant expression of BFP (Fig. 4d). GD2-BFP T cells developed with wild-type GD2 scFv showed significantly lower BFP activation in comparison to the high-affinity GD2 scFv-containing GD2-BFP T cells. Further optimizing the spacer between scFv and Notch core regions of the constructs with either CH2–CH3 or hinge domain of IgG4 in the GD2-SynNotch system did not improve the gate activity of GD2$^{WT}$ compared to GD2$^{E101K}$ SynNotch receptor (Supplementary Figs. 1c and 5b). We next constructed a GD2$^{E101K}$ SynNotch receptor as the gate for the B7H3 CAR (GD2-B7H3) (Fig. 4e). GD2-B7H3 T cells showed significant upregulation of CD107a only in response to GD2$^+$B7H3$^+$ (CHLA255) and not to GD2$^-$B7H3$^+$ (LAN6) NBL cells (Fig. 4f). GD2-B7H3 T cells had significant expansion and demonstrated secretion of known effector cytokines in the presence of NBL cells (Fig. 4g, h). GD2-B7H3 T cells, but not UT cells, also showed potent cytotoxicity against CHLA255 in vitro (Fig. 4i). While GD2-B7H3 T cells showed significant NBL cytotoxicity at T cell to tumor ratios of 5:1 or greater, there was a longer time before control of in vitro growth at the lowest T cell to tumor cell ratio of 2:1 compared to constitutively expressing B7H3 CAR-T cells

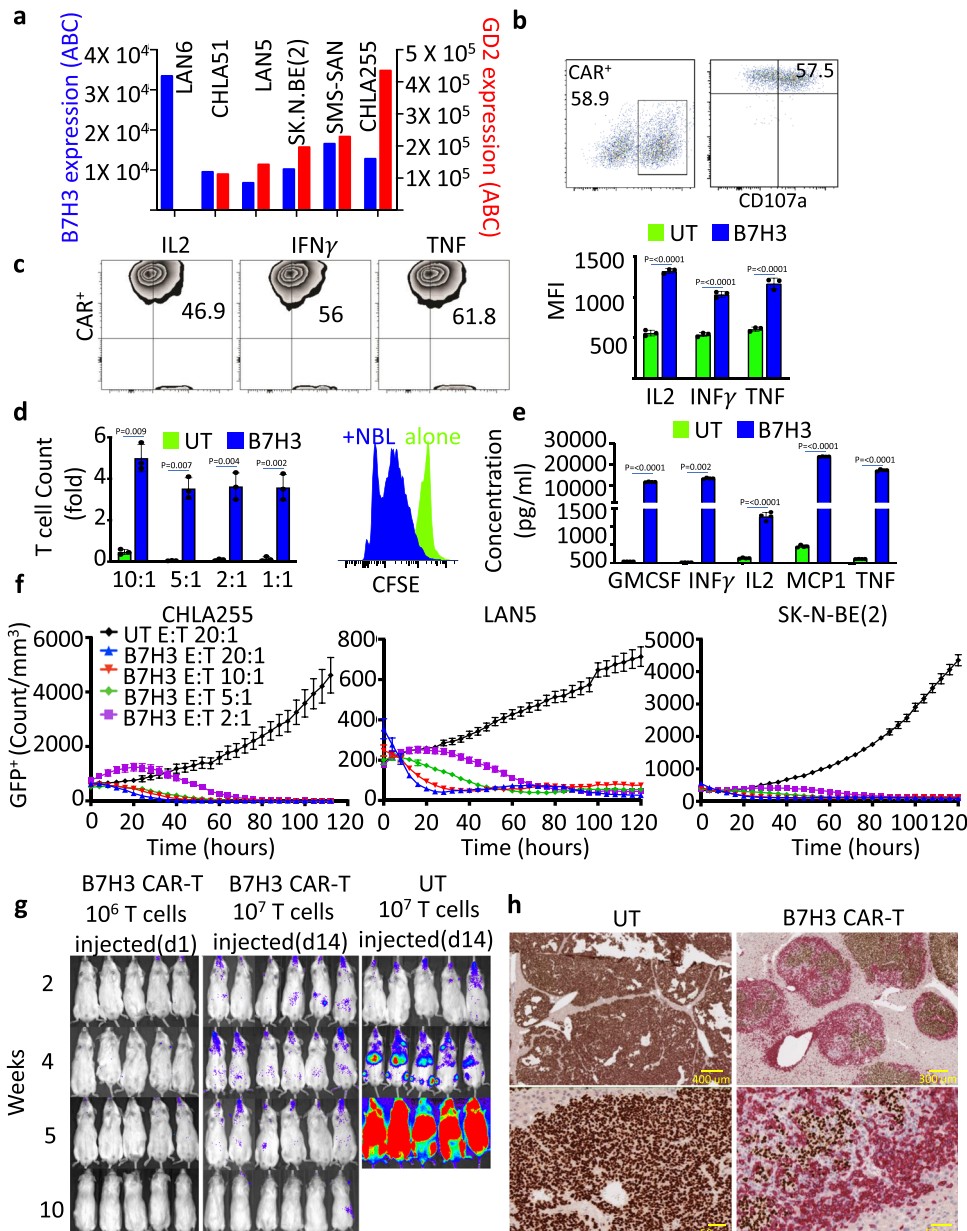

**Fig. 3 B7H3 CAR-T cells show effective anti-tumor activity in several NBL models. a** B7H3 and GD2 expression on six human NBL cell lines using Quantibrite™ beads. **b** Enumeration by flow cytometry of (left) human T cells transduced with lentivirus to express B7H3-BBz construct and (Right) the activation marker CD107a on transduced T cells (CD3) co-cultured with the NBL cell line CHLA255 for 4 h. **c**, left: Representative contour plot of intracellular cytokines IL2, IFNγ, and TNF in B7H3 CAR-T cells co-cultured with NBL cells for 48 h; (right) summary of the data as measured by flow cytometry. **d**, left: Expansion of UT and B7H3 CAR-T cells co-cultured for 5 days with CHLA255 NBL cells at various effector:target ratios. Right: Histogram of CFSE incorporation demonstrating the proliferation of B7H3 CAR-T cells only when co-cultured with NBL cells. **e** Summary of cytokine release of GM-CSF, IFNγ, IL2, MCP1, and TNF measured by ELISA from UT and B7H3 CAR-T cells co-cultured with CHLA255 NBL cells at 1:1 effector:target ratio. **f** Kinetics of mean cytotoxicity of UT and B7H3 CAR-T using live-cell imaging (enumeration of GFP⁺ tumor cells) against NBL cell lines (CHLA255, LAN5, and SK-N-BE (2)) at indicated E:T ratios. **g** Representative bioluminescence images of CHLA255 tumor growth upon intravenous (i.v.) injection into NSG mice (1× $10^6$ cells/mouse) and treatment 14 days later with i.v. injection of B7H3 CAR-T or UT cells (1 × $10^7$ on day 14). Surviving mice were followed for a minimum of 100 days post-tumor inoculation. **h** Representative images from immunohistochemical analysis of liver tissue obtained from mice injected i.v. with 1× $10^6$ CHLA255 followed by i.v. injection of 1 × $10^7$ UT cells (left) or 1 × $10^7$ B7H3 CAR-T cells (right) on day 28 (high disease burden). Tissues were obtained 7 days post T-cell injection and stained for NBL-specific marker PHOX2b (brown) and human CD3 (red). Despite mice not surviving with CAR-T cells at this high disease burden stage, impressive T-cell infiltration and tumor reduction were identified. Data shown are representative of three (**b**, **c**) and four (**e**) independent experiments. minimum of four replicate (**f**), mean ± SD (**c–f**). Two-tailed t test (**c–e**) n = 5 mice (B7H3 d14, B7H3 d1, and UT), n = 3 mice (high burden, **h**). Individual BLI and source data are provided as a Source Data file.

(Figs. 4i and 3f). GD2-B7H3 T cells were next tested in vivo in a metastatic NBL xenogeneic model in which infusion of 1 × $10^7$ GD2-B7H3 T cells (estimated based on the fraction of GD2 scFv × fraction of B7H3 scFv as detected by flow cytometry)

on day 7 post-tumor inoculation led to complete cure. In contrast, mice receiving control CD19-B7H3 T or UT cells similarly developed tumors (Fig. 4j). Histologic evaluation of liver tissue from metastatic NBL mice with well-established tumors following

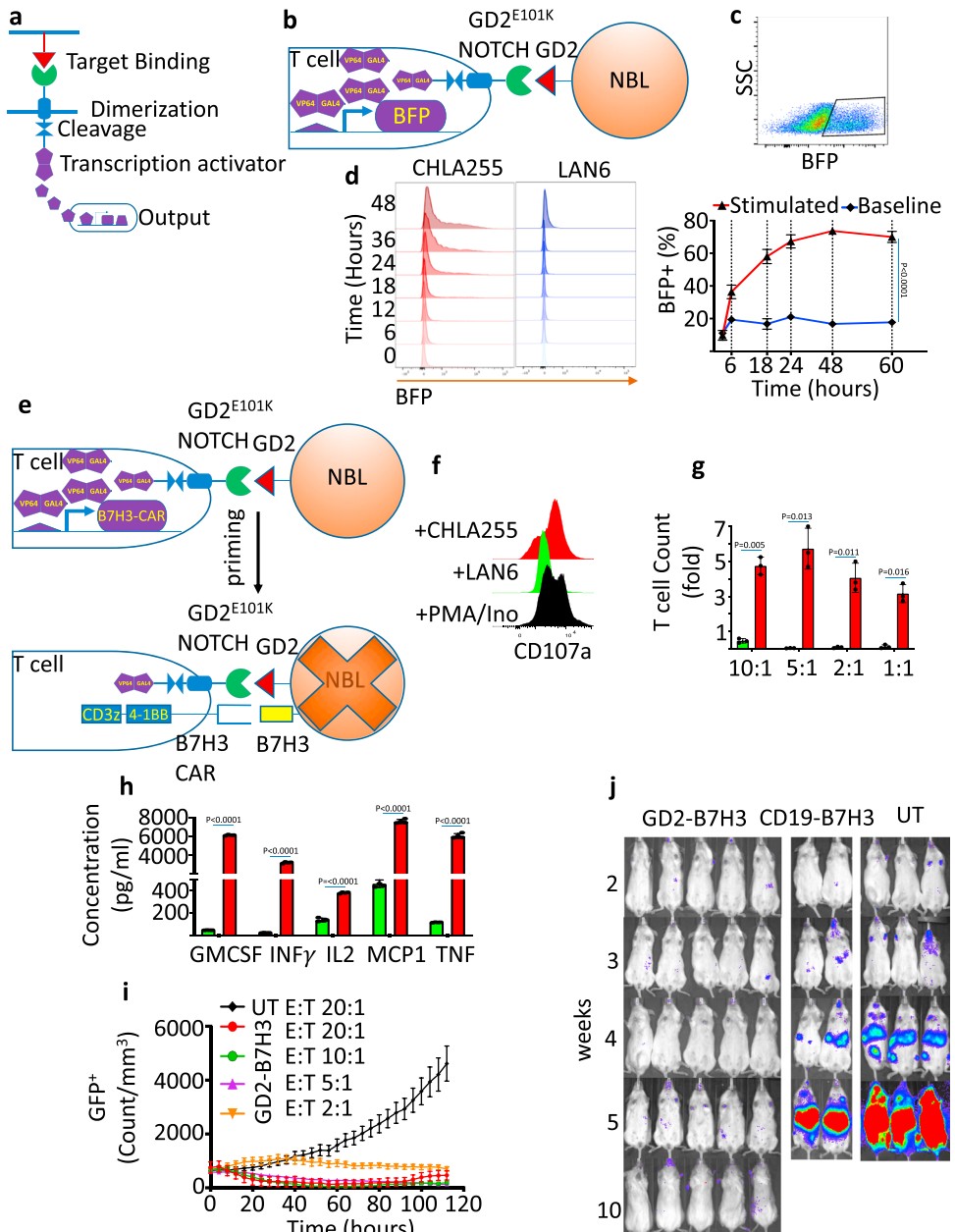

**Fig. 4 Construction of highly specific and effective GD2-B7H3 T cells. a** Schematic of SynNotch receptor with extracellular ligand-binding domain (scFv) directed against a TAA. Upon ligand recognition by the SynNotch receptor, an orthogonal transcription factor is cleaved from the cytoplasmic tail that activates a custom genetic construct. **b** Schematic structure of activation of the BFP reporter gene when the high-affinity scFv GD2^E101K SynNotch receptor is presented with GD2 TAA. **c**, upper: Representative dot plot of BFP expression by flow cytometry in GD2-BFP T cells co-cultured with NBL cells for 24h. Lower: Time-course kinetics of BFP expression by flow cytometry in GD2-BFP T cells with and without NBL co-culture for 3 days. **d** Histograms of BFP expression in GD2-BFP T cells co-cultured with GD2^+ CHLA255 and GD2^− LAN6 NBL cells from 0 to 48 h. **e** Schematic structure of B7H3 CAR activation when the high-affinity scFv GD2^E101K SynNotch receptor is presented with GD2 TAA and subsequent expression of B7H3 CAR and resultant cytotoxic activity. **f** Histogram representing the activation marker (CD107a) on GD2-B7H3 cells co-cultured with GD2^+ NBL cell line CHLA255, GD2^− LAN6, or PMA/Ionomycin (positive control) after 24h. **g** Expansion of UT, GD2-B7H3 T cells co-cultured for 5 days with CHLA255 NBL cells at various effector:target ratios. **h** Summary of cytokine release of GM-CSF, IFNγ, IL2, MCP1, and TNF measured by ELISA from UT and GD2-B7H3 T cells co-cultured with CHLA255 NBL cells at 1:1 effector:target ratio **i** kinetics of cytotoxicity of UT and GD2-B7H3 T using live-cell imaging (enumeration of GFP^+ tumor cells) against CHLA255 NBL cells at indicated E:T ratios. **j** Representative bioluminescence images of CHLA255 NBL tumor growth upon i.v. injection into NSG mice (1×10^6 cells/mouse) and treatment 7 days later with i.v. injection of 1×10^7 UT, GD2-B7H3, or CD19-B7H3 T cells. Surviving mice were followed for a minimum of 100 days post-tumor inoculation. Data shown are representative of three (**c**, **d**, **f**, **g**) and four (**h**) independent experiments. minimum of four replicate (**i**), mean ± SD (**c**, **g**, **h**, **i**). Two-tailed *t* test (**c**, **g**, **h**). n=5 mice (GD2-B7H3, CD19-B7H3, and UT). Individual BLI and source data are provided as a Source Data file.

treatment with $1 \times 10^7$ GD2-B7H3 T cells showed infiltration of T cells in the tumor bed (Supplementary Fig. 6), although the extent of penetration was less than that seen following conventional B7H3 CAR-T treatment. Overall, these optimized GD2-

B7H3 T cells demonstrated excellent in vitro and in vivo specificity and efficacy against GD2^+B7H3^+ NBL cell lines.

To accurately evaluate the metabolic function and efficacy of GD2-B7H3 T cells, we modified the response vector by adding a

truncated version of human CD19 (tCD19) under its own promoter. This allowed us to enrich for GD2-B7H3 T cells using anti-CD19 beads to ensure a high frequency of double-positive T cells in our analyses (see "Methods", Fig. 5a, b, and Supplementary Fig. 7a). The tCD19 could also act as a safety switch for future clinical trials as it can be targeted by Blinatumomab, a CD19 bispecific T-cell engager (BiTE). The enriched GD2-B7H3 T cells upregulated the B7H3 CAR, CD69 activation marker, and intracellular cytokines only in the presence of double-positive target cells and showed improved anti-tumor effect at lower E:T ratios (Fig. 5c and Supplementary Fig. 7b–d). In vitro, enriched GD2-B7H3 cells also showed high specificity against cell lines expressing both B7H3 and GD2 (CHLA255, LAN6$^{GD2+}$) but not the wild-type LAN6 expressing B7H3 only (Fig. 5d, e). In vivo, enriched GD2-B7H3 T cells also showed anti-tumor activity in mice bearing LAN6$^{GD2+}$ but not wild-type LAN6 tumors (Fig. 5f). Enriched GD2-B7H3 T cells but not CD123-B7H3 showed a high cure rate in mice with metastatic GD2$^+$B7H3$^+$ CHLA255 NBL (Fig. 5g) and importantly showed considerable CD3 expansion in peripheral blood compared to conventional B7H3 T cells (Fig. 5h). However, the enrichment process did not show any effect on mice treated on day 14 after tumor inoculation (Fig. 5i and Supplementary Fig. 8a, b). We noticed that enriched CD123-B7H3 T cells also showed in vitro cytotoxicity at the highest T cell to tumor ratio of 20:1, suggesting a low level of leaky expression of B7H3 CAR (Supplementary Fig. 9a); however, this did not translate to an in vivo impact and did not result in T-cell trafficking at disease sites (Fig. 5g and Supplementary Fig. 9b). As the gated system should have inherent safety due to lack of significant activation until an encounter with the first TAA, we elected to test if repeated infusions would cause toxicity or potentially improve late timepoint treatment efficacy. GD2-B7H3 T cells infused every 72 h for three doses starting day 14 post-tumor inoculation resulted in significantly enhanced anti-tumor effect and doubled survival time. Interestingly, the majority of failures in animals treated with multiple infusions occurred in the brain and bone marrow in contrast to those treated with a single dose of GD2-B7H3 (Fig. 5i and Supplementary Fig. 8a, b), suggesting slower cytotoxicity kinetics may contribute to its lower efficacy in high tumor burden models. These data suggest that the construction of highly enriched and specific GD2-B7H3 T cells is possible, and efficacy can be improved with repeated infusions.

**GD2-B7H3 T cells are metabolically fit and resilient to exhaustion**. Next, we explored the expression of cell surface exhaustion markers in our enriched gated system versus conventional B7H3 CAR-T cells. Expression of exhaustion markers PD1 and LAG3 was significantly lower in GD2-B7H3 T cells compared to B7H3 CAR-T cells exposed to NBL cells (Fig. 6a). GD2-B7H3 T cells also retained higher levels of the activation markers CD27 and CD25 (Fig. 6b). The oxygen consumption rate (OCR) of UT, GD2-B7H3 T, and B7H3 CAR-T cells after 48 h of co-incubation with NBL cells was measured through serial additions of oligomycin (an inhibitor of ATP synthesis), carbonyl cyanide-p trifluoromethoxyphenylhydrazone (FCCP; an uncoupling ionophore), and rotenone with antimycin A (blocking agents for complexes I and III of the electron transport chain, respectively) to discern the relative contributions of the mitochondrial and non-mitochondrial mechanism of oxygen consumption. B7H3 CAR-T cells demonstrated higher basal OCR post-stimulation, while GD2-B7H3 T cells showed a similar profile as UT cells (Fig. 6c). These data demonstrate a significantly higher respiratory capacity in GD2-B7H3 T cells compared to B7H3 CAR-T cells after neuroblastoma-directed cytotoxic activity that is on par with naïve T cells (Fig. 6c). Gene

Set Enrichment Analysis (GSEA) of the RNA expression profile of T cells obtained 48 h post co-culture with target cells demonstrated enrichment of genes with memory T-cell phenotype in B7H3 CAR-T cells compared to GD2-B7H3 T cells (Fig. 6d). However, B7H3 CAR-T cells were enriched in genes associated with glycolysis and apoptosis pathways, suggesting a committed terminal fate for these T cells (Fig. 6d). There was significantly greater expression of genes attributed to exhaustion in B7H3 CAR-T cells compared to GD2-B7H3 T cells (Fig. 6e). In particular, B7H3 CAR-T cells showed higher expression of inhibitory receptors LAG3, HAVCR2 (TIM3), and BTLA genes, along with exhaustion-related transcription factor genes TBX21 (T-bet), PRDM1 (Blimp-1), and IKZF2 (Helios). B7H3 CAR-T cells also expressed genes that encode transcription factors reported to be associated with activated and memory T cells, such as KLF6, JUN, and JUNB. These results suggest that reductions in some exhaustion-associated genes but also T-cell activation-associated genes in GD2-B7H3 T cells after tumor-killing differentiates these seemingly naive T cells from their exhausted and pre-apoptotic conventional counterpart. To determine if the resilience of gated T cells to exhaustion translates to any in vivo advantage, we first continuously co-cultured GD2-B7H3 T cells and B7H3 CAR-T with CHLA255 cells for 12 days adding fresh tumor cells to the culture every 48 h. The GD2-B7H3 T cells chronically exposed to NBL cells showed lower expression of exhaustion markers PD1, TIM3, and LAG3 compared to similarly exposed B7H3 CAR-T cells (Fig. 6f). Importantly, the chronically exposed GD2-B7H3 T cells showed significantly higher peripheral blood expansion in metastatic NBL-bearing mice and were able to prevent tumor progression in three out of four mice, while the chronically exposed B7H3 CAR-T cells failed to expand in vivo and were unable to prevent tumor formation in any of the mice (Fig. 6g, h). Overall, our data suggest that GD2-B7H3 T cells are metabolically distinct from B7H3 CAR-T cells, with the former displaying a greater capacity for oxidative metabolism and lower propensity for the exhaustion that contributes to prolonged persistence and proliferation in blood and anti-tumor efficacy.

**B7H3-GD2 T cells penetrate CNS but do not cause neurotoxicity**. To address the safety of GD2$^{E101K}$ as a CAR in a gated model, we constructed B7H3-GD2 T cells, where the B7H3 SynNotch receptor controls the expression of the GD2$^{E101K}$ CAR (Fig. 7a and Supplementary Fig. 1b). We validated the B7H3 as a gate construct by demonstrating BFP activation in B7H3-BFP T cells when co-cultured with the CHLA255 NBL cell line (Fig. 7b and Supplementary Fig. 10a, b). B7H3-GD2 T cells, enriched using tCD19, exhibited in vitro cytotoxicity against CHLA255 similar to GD2-B7H3 T cells (Fig. 7c). B7H3-GD2 T cells but not UT cells prevented disease progression in xenografted metastatic NBL mice, and none of the mice exhibited evidence of neurotoxicity (Fig. 7d). However, single-cell analysis of brain tissue of mice euthanized 7 weeks post T-cell infusion demonstrated the presence of T cells in mice treated with B7H3-GD2 CAR-T cells but not those treated with GD2-B7H3 (Fig. 7e, f, and Supplementary Figs. 11, 12). These data show that B7H3-GD2 T cells can avoid neurotoxicity; however, a portion of likely activated B7H3-GD2 T cells can still infiltrate the CNS parenchyma.

**Discussion**
The promise of CAR-T-cell therapy in solid tumors has been stifled by difficulties in optimizing on-target efficacy while better understanding and minimizing off-target toxicity. GD2 CAR-T-cell-associated neurotoxicity in murine models had only been described with the use of high-affinity GD2 scFv; however, herein,

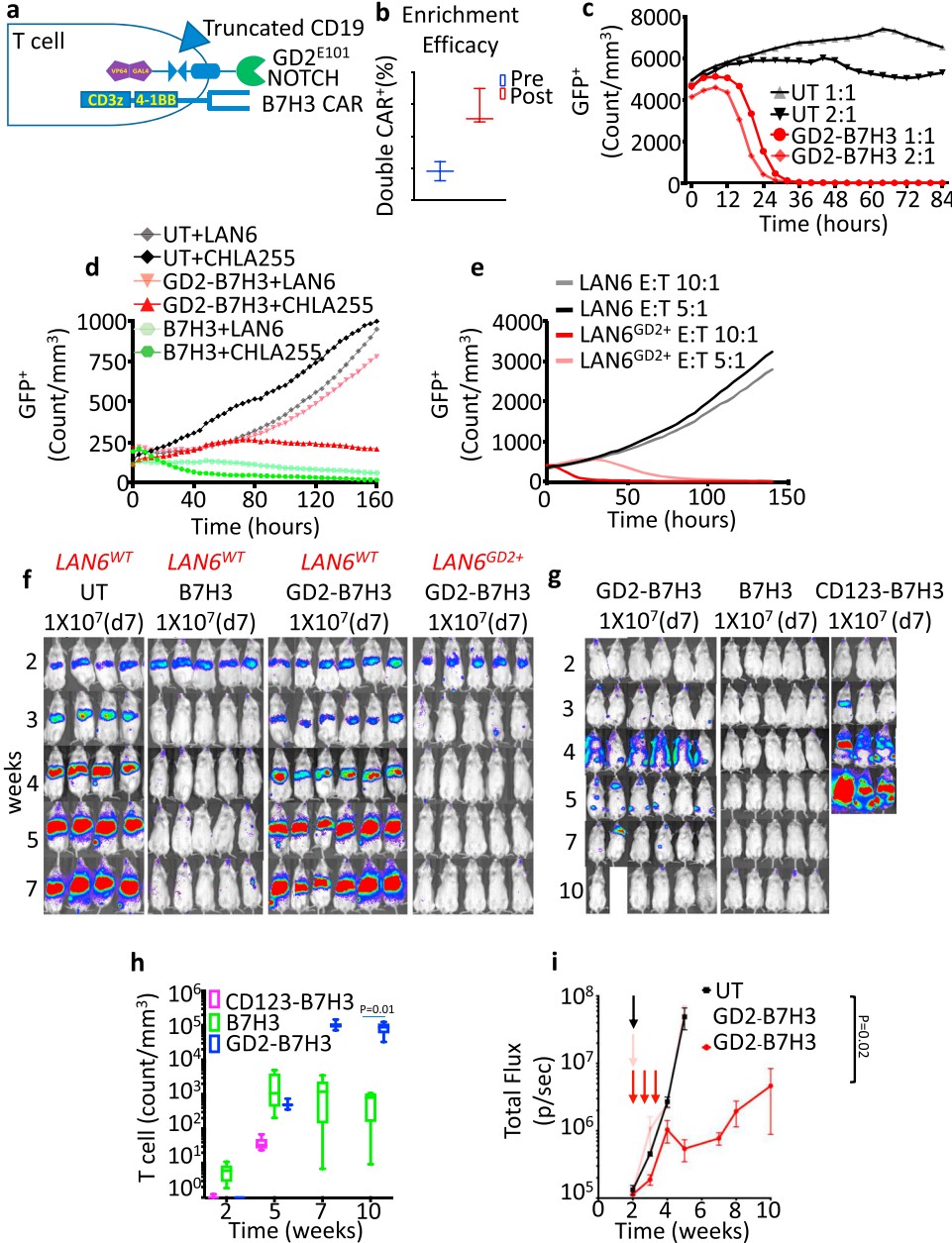

**Fig. 5 Enrichment of GD2-B7H3 T cells improves efficacy against NBL. a** Schematic of GD2-B7H3 T cells engineered to constitutively express truncated CD19. Double-positive T cells for SynNotch and CAR constructs can be enriched post transduction via CD19 beads and anti-Fab beads to achieve a higher yield. **b** Average and range of double-positive (CD19, GD2 SynNotch) GD2-B7H3 T cells pre- and post-enrichment as measured by flow cytometry. **c** Kinetics of cytotoxicity of UT and enriched GD2-B7H3 T cells using live-cell imaging (enumeration of GFP+ tumor cells) against CHLA255 cells at indicated E:T ratios. **d** Kinetics of cytotoxicity of UT, B7H3 CAR-T, and enriched GD2-B7H3 T cells using live-cell imaging (enumeration of GFP+ tumor cells) against GD2+ CHLA255 and GD2- LAN6 cells at 1:1 E:T ratio. **e** Kinetics of cytotoxicity of GD2-B7H3 T cells using live-cell imaging (enumeration of GFP+ tumor cells) against wild-type LAN6 (GD2 negative) and LAN6$^{GD2+}$ cell lines at indicated E:T ratios. **f** Bioluminescence images of LAN6 and LAN6$^{GD2+}$ metastatic tumor-bearing mice treated with i.v. injection of $1 \times 10^7$ enriched GD2-B7H3, UT, or B7H3 CAR-T cells. Surviving mice were followed for a minimum of 100 days after injection of the NBL cell line. **g** Bioluminescence images of CHLA255 tumor-bearing mice treated with i.v. injection of $1 \times 10^7$ enriched GD2-B7H3, CD123-B7H3, or B7H3 CAR-T cells 7 days post NBL cell line injection. Surviving mice were followed for a minimum of 100 days post-tumor inoculation. **h** Boxplot data of peripheral blood T-cell count over time in mice (from Fig. 4g) treated with enriched GD2-B7H3, CD123-B7H3, or B7H3 CAR-T cells. **i** Summary bioluminescence intensity data from CHLA255 tumor-bearing mice (Supplementary Fig. 8) treated 14 days after tumor inoculation with three doses of i.v. injection of UT or enriched GD2-B7H3 T cells ($1 \times 10^7$ cells/mouse/injection) versus one-time i.v. injection of GD2-B7H3 T cells, mean for a minimum of four replicate (**c–e**), min to max (**b**). Whiskers were calculated using the Tukey method. Boxes extend from 25th to 75th percentiles, and lines in the middle of the boxes represent the median (**h**), mean ± SD (**i**). Two-tailed $t$ test (**h**, **i**). $n = 6$ mice (GD2-B7H3 (LAN6, CHLA255)), $n = 5$ (B7H3(LAN6, CHLA255)), $n = 4$ (UT), $n = 3$ (CD123-B7H3). Individual BLI and source data are provided as a Source Data file.

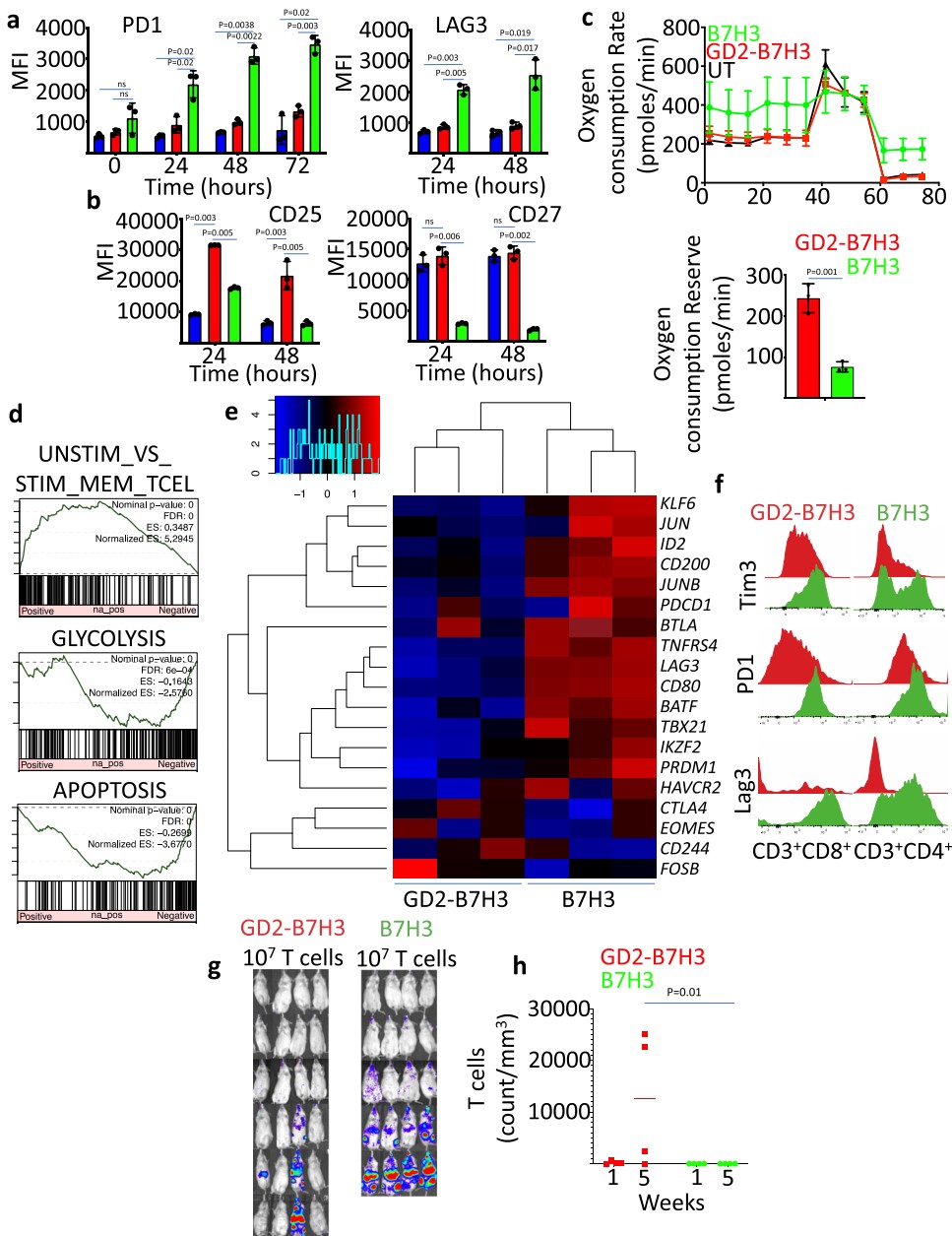

**Fig. 6 GD2-B7H3 are metabolically fit and resilient to in vivo and in vitro exhaustion. a** Summary of the data representing expression of exhaustion markers PD1 and LAG3 enumerated by flow cytometry in UT, B7H3 CAR-T cells, GD2-B7H3 T cells co-cultured with CHLA255 NBL cells at indicated times. **b** Summary of the data representing expression of activation markers CD25 and CD27 enumerated by flow cytometry in UT, B7H3 CAR-T cells, and GD2-B7H3 gated CAR-T cells co-cultured with CHLA255 NBL cells at indicated times. **c**, left: Oxygen consumption rate (OCR) as measured by Seahorse assay in UT, B7H3 CAR-T cells, and GD2-B7H3 gated CAR-T cells co-cultured with CHLA255 NBL cells for 48 h. Right: Summary of the data representing oxygen consumption reserve in GD2-B7H3 T cells and B7H3 CAR-T cells. **d** GSEA enrichment plot of significantly ranked pathways (MsigDB C5 gene ontology) from RNA sequencing data of GD2-B7H3 T cells and B7H3 CAR-T cells co-cultured with CHLA255 NBL cells for 3 days (tumor cells are eliminated in either group after co-culture). **e** Heatmap of gene expression of exhaustion-related transcription factors (*TBX21*, *EOMES*, *PRDM1*, *IKZF2*), inhibitory receptors (*LAG3*, *HAVCR2*, *CTLA4*, *BTLA*, *CD244*), and transcription factors reported being preferentially expressed in memory T cells (*KLF6*, *JUN*, *JUNB*) from RNA sequencing data of GD2-B7H3 T cells and B7H3 CAR-T cells isolated post co-culture with CHLA255 NBL cells for 3 days (tumor cells are eliminated in either group after co-culture). **f** Histogram of PD1, LAG3, TIM3 protein expression in T cells co-cultured repeatedly with CHLA255 cells for 12 days. **g** Bioluminescence images of CHLA255 tumor-bearing mice treated 1 day after tumor injection with i.v. injection of 1 × 10$^7$ GD2-B7H3 or B7H3 CAR-T cells previously co-cultured repeatedly with CHLA255 cells for 12 days. Surviving mice were followed for a minimum of 100 days post-tumor inoculation. **h** Summary data representing CAR-T-cell count in peripheral blood of animals in Fig. 5g at week 1 and week 5 post CAR-T-cell infusion. Data shown are representative of three independent experiments (**a**–**e**), mean ± SD (**a**–**c**). Two-tailed *t* test (**a**–**c**, **h**). *n* = 4 mice (GD2-B7H3, B7H3). Data shown are representative of three individual mice, remaining images are included in the Supplementary Information (**e**). Individual BLI and source data are provided as a Source Data file.

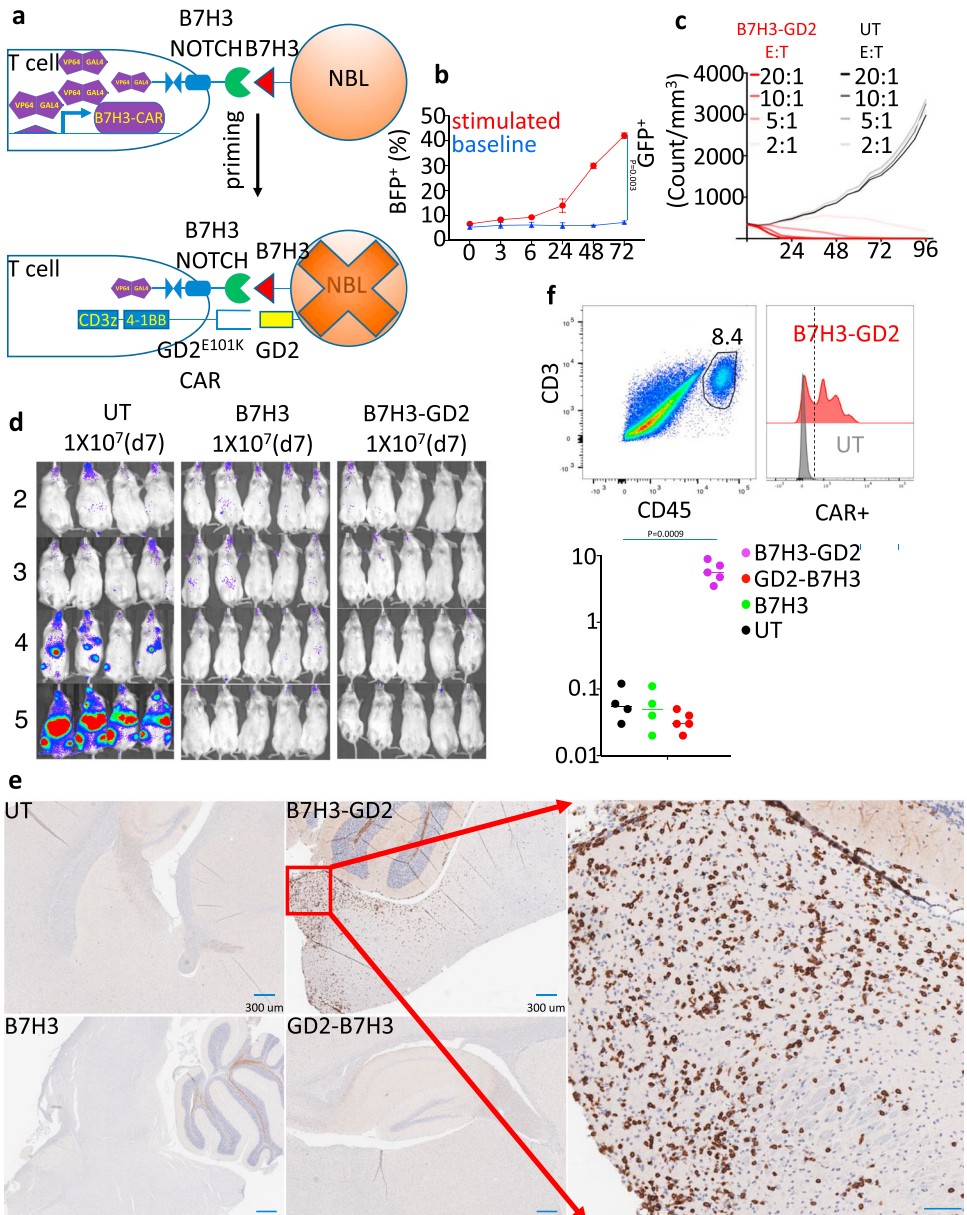

**Fig. 7 B7H3-GD2 T cells penetrate CNS but do not cause neurotoxicity. a** Schematic of SynNotch receptor with extracellular ligand-binding domain (scFv) directed against B7H3. Upon ligand recognition by the SynNotch receptor, an orthogonal transcription factor is cleaved from the cytoplasmic tail that activates GD2-BBz CAR (101-GD2 which uses GD2E101K scFv). **b** Time-course kinetics of BFP expression by flow cytometry in B7H3-BFP T cells co-cultured with B7H3+ CHLA255 NBL for 3 days. **c** Kinetics of cytotoxicity of UT and B7H3-GD2 T using live-cell imaging (enumeration of GFP+ tumor cells) against CHLA255 NBL cells at indicated E:T ratios. **d** Bioluminescence images of CHLA255 NBL tumor growth upon i.v. injection into NSG mice ($1 \times 10^6$ cells/mouse) and treatment 7 days later with i.v. injection of $1 \times 10^7$ of UT, B7H3, and B7H3-GD2 T cells. Surviving mice were followed for a minimum of 100 days post-tumor inoculation. **e** Immunohistochemical analysis of CD3 expression (brown staining) in the brain of mice treated with UT, B7H3, B7H3-GD2, and GD2-B7H3 CAR-T cells. **f** left: Representative flow cytometry plots from single-cell homogenized brain tissue stained for murine CD45 and CD3, (middle) histogram of CAR-T cells identified among infiltrating CD3+ T cells using an anti-Fab antibody. Right: Summary data representing the percentage of T cells in total live single cells in the brain of treated animals. Data shown are representative of three independent experiments (**b**), mean for minimum of four replicate (**c**), mean ± SD (**b**). Two-tailed t test (**b**, **f**). n = 5 mice (B7H3-GD2, B7H3) and n = 4 mice (UT). Additional immunohistochemical images are provided in the Supplementary Information. Individual BLI and source data are provided as a Source Data file.

we demonstrate the novel finding that neurotoxicity can also occur with wild-type GD2 scFv, but it is associated with the costimulatory domain used in CAR construction. The unpredictable off-tumor toxicity associated with GD2 CAR-T cells, combined with our work demonstrating impressive in vitro and in vivo anti-NBL efficacy of a B7H3 CAR, led us to develop SynNotch-gated GD2 and B7H3 CAR-T cells, which have improved specificity and avoid neurotoxicity while retaining the

ability to control tumor growth and cure metastatic NBL. The GD2-B7H3 T cells exhibited greater metabolic fitness, lower exhaustion profile, and superior in vivo anti-tumor efficacy after repeated in vitro stimulation compared to conventional B7H3 CAR-T cells.

B7H3, a pan-cancer antigen, is broadly expressed on many pediatric solid tumors, including NBL[20,28]. Recently, B7H3 CAR-T cells demonstrated potent activity against xenograft models of

pediatric cancers and adult ovarian carcinoma and pancreatic tumors[20,25]. A murine version of B7H3 CAR-T cells was developed by others to assess safety in an immunocompetent mouse model and did not exhibit off-tumor toxicity, which is encouraging[25]. However, inferences about off-tumor toxicity in human models should be considered with caution as the murine B7H3 CAR-T cells used in these experiments were cytotoxic only to cells where murine B7H3 was forcibly expressed and not to those with endogenous expression of B7H3[25]. In addition, differences between human and murine B7H3 expression patterns exist, and vigilance should be exercised to ameliorate potential human toxicities. While there is 87% homology between murine and human B7H3, the pattern of expression in various tissues is not identical[29]. Yan et al. demonstrated B7H3 expression only in murine bladder epithelial cells, while Du et al. found positive expression in the murine stomach, small intestine, and colon, which could be explained by the use of different antibodies[25,29]. Indeed, expression patterns of B7H3 in human samples also differ by the antibody used, with consistent expression observed in the placenta for all antibodies tested, while expression at sites such as the pancreas and liver is antibody-dependent, and all of these sites are not identified in mouse studies of B7H3 expression. It is also difficult to ascertain if low B7H3 expression could spare normal tissue from CAR-T-cell off-tumor toxicity as the antigen threshold for CAR-T-cell toxicity is not well defined. For example, significant pulmonary toxicity was observed with HER2 CAR-T cells despite known low-level expression of this antigen on the lung epithelium, suggesting that antigen expression on non-tumor tissues may not correlate with the risk of toxicity in those tissues[3,30]. Given the novelty of CAR-T-cell therapy in solid tumors, toxicity potential is often inferred from toxicity data obtained from antibody therapies directed at the same TAA, but the ability to extrapolate from these studies and accurately predict CAR-T-cell toxicity, even when utilizing the scFv of the same antibody, is not yet clear. For example, antibodies against GD2 have a long track record of clinical use in children with NBL, yet the significant universal pain associated with anti-GD2 antibody infusion has not been observed in clinical trials of GD2 CAR-T cells. Further studies on inferences about toxicity from murine CAR-T-cell studies or antibody use in human clinical trials remain challenging and fraught with uncertainty.

The neurotoxicity observed in mice treated with murine GD2-28z but not GD2-BBz CAR-T cells demonstrates that in the presence of the same scFv, the choice of costimulatory signaling domain can significantly influence CAR-T-cell off-tumor toxicity. CAR design has been shown to affect CAR-T-cell function and toxicity in a clinically significant manner[31–33]. The lower incidence of neurotoxicity observed in a CD19 CAR-T-cell trial was hypothesized to be associated with the structural characteristics involving the choice of the hinge and transmembrane region of the CD28 costimulatory domain. CD19 CAR-T cells utilizing the native CD28 but not the CD8 hinge and transmembrane regions exhibited higher cytokine production in vitro and led to a higher level of neurotoxicity in vivo. It has been suggested that the tendency for self-dimerization at the CD28 transmembrane drives the higher intensity of CAR activation and leads to its toxicity[34,35]. CAR-T cells directed against GD2 have also shown variable efficacy and toxicity in preclinical studies depending on their structural characteristics[18,36–39]. Fatal neurotoxicity has been observed in studies with GD2$^{E101K}$ CAR-T cells harboring a 4-1BB costimulatory domain[18,38], which is the same design used in our B7H3-GD2$^{E101K}$ CAR construct. In contrast, Lynn et al. showed no evidence of neurotoxicity from a GD2$^{E101K}$ CAR-T cell using CD28z costimulatory domain with a CH2–CH3 linker[37]. Other preclinical studies using GD2$^{wt}$ CAR-T cells have also not demonstrated neurotoxicity[18,36]. While the mechanism

of neurotoxicity remains unclear, our data demonstrate that only certain GD2 CAR constructs can infiltrate the CNS. While cytokine release can be a factor in observed neurotoxicity as previously suggested[40], our data indicate a role for on-target off-tumor toxicity. Further research is required to understand the differential toxicities observed in certain GD2 CARs and the effects of their structural design on downstream signaling. Our findings support and extend the concept that structural features in CAR construction, including scFv binding affinity, choice of hinge molecules, and costimulatory domains, can alter CAR-T-cell toxicity. SynNotch and other inducible design approaches that aim to reduce off-tumor toxicity while retaining anti-tumor efficacy should be studied and considered as additional tools to safeguard against unwanted toxicity in treating human solid tumors.

The SynNotch system was proposed to minimize off-tumor toxicity of CAR-T cells, relying on an "AND" logic gate model to regulate the state of T-cell activation and effector function[9,41,42]. The SynNotch approach for combinatorial antigen recognition is a critical advancement in T-cell therapies as most other combinatorial antigen recognition strategies rely on integrating signaling from multiple partially functional CARs (split CARs) that work cooperatively or antagonistically to control the activation of T cells[9,41,42]. The SynNotch system, unlike split CARs, spatially and temporally separates activation signals, which eliminates the risk of partial activation. Post activation and without the presence of target cells, the gated CAR expression decays over time. The decay kinetics, along with the number of circulating cells expressing CAR, likely determines the off-tumor toxicity of SynNotch-gated CAR-T cells. The lack of toxicity seen in our B7H3-GD2 T-cell-treated mice, despite their presence in brain tissue with GD2$^{E101K}$ CARs, could be attributed to the decay of CAR expression post CNS infiltration and likely to a lower number of infiltrating T cells. Additional studies will be required to fully assess the mechanism affecting this temporal decay and the role of tumor burden on off-tumor toxicity in these gated CAR-T cells. Our choice of GD2 as the gatekeeper for B7H3 CAR expression was based on the presence of GD2 in the brain and the desire to eliminate any possibility of CNS toxicity. The GD2-B7H3 T cells also proved highly specific in vitro and in vivo, reliably targeting GD2- and B7H3-expressing NBL and not recognizing the rare NBL cell lines that lack GD2 expression. An additional limitation of the current synNotch system is the reliance on a non-human orthogonal transcription factor that could be immunogenic, and alternative transcription factors or structure-guided deimmunization may be necessary for clinical application.

GD2-B7H3 T cells exhibited superior resistance to exhaustion and greater metabolic fitness in comparison to conventional CAR-T cells. Metabolic preference (glycolytic versus oxidative) has an enormous impact on T-cell fate[43–45]. The success of 4-1BB CAR-T cells is thought to be partly attributable to improved fitness with the use of oxidative phosphorylation to generate ATP and enhance persistence[46]. After eradicating NBL cells, GD2-B7H3 T cells had an oxygen consumption rate similar to UT cells, suggesting that the gated T cells can revert to their naïve metabolic state. Improved metabolic plasticity and reprogramming in favor of oxidative rather than glycolytic phosphorylation supports the hypothesis that the gated CAR-T cells likely have intact expansion potential, similar to unmanipulated naïve T cells. We observed significant enrichment of glycolytic genes in conventional B7H3 CAR-T cells compared to GD2-B7H3 T cells, supporting our metabolic studies. Our finding does not contradict a previous publication that demonstrated oxidative metabolism in 4-1BB CAR-T cells[46] as this comparison was made in contrast to the highly glycolytic 28z CAR-T cells. Our data suggest that

SynNotch-gated CAR-T cells have a more favorable oxidative metabolic profile compared to 4-1BB CAR-T cells. We attribute these findings to the high metabolic fitness of the gated CAR-T cells and in line with the observation that gated CAR-T cells, in contrast to conventional CAR-T cells, do not exhibit a pre-apoptotic gene signature.

Recently, Srivastava et al.[41] demonstrated the discriminatory power of SynNotch CAR-T cells against normal tissue expressing ROR1. Our work reinforces and expands those findings beyond proof of concept in a different, clinically relevant disease model, but unlike ROR1 SynNotch CARs, GD2-B7H3 T cells show significant efficacy and survival benefits in a very aggressive tumor model. GD2-B7H3 T cells also had improved expansion in vivo and the ability to maintain a lower exhaustion profile as compared to conventional B7H3 CAR-T cells. Importantly, when CARs were exposed for long periods to tumor cells in vitro and subsequently used for in vivo experiments, the GD2-B7H3 T cells demonstrated superior anti-tumor efficacy to conventional B7H3 CARs. The lower in vivo anti-tumor efficacy observed for GD2-B7H3 T cells against animals with high-burden tumors may be due to their lower cytotoxic efficiency as demonstrated in vitro with longer times required for gated CAR to achieve a similar level of cytotoxicity compared to their conventional counterpart. The lower efficiency may be due in part to slower kinetics resulting from anti-B7H3 CAR proteins that cannot be expressed until engagement with the initial GD2 antigen. Increasing the number of gated CARs did improve in vivo efficacy, as noted with repeat infusions of gated CAR-T cells. Our data suggest altered binding affinity of scFv used in the SynNotch receptor can also enhance CAR expression, and further efforts should be directed at methods that decrease the time to CAR expression and increase the number of expressed proteins after engagement of T cells with the first TAA. Given the favorable safety profile of gated CAR-T cells and our current data, it is likely that efficacy can be improved through multiple infusions of gated CAR-T cells. A potential escape mechanism or consequence of improved efficacy may be lost or diminishing density of tumor antigen, as has been demonstrated for CAR-T-cell strategies directed against CD19 or CD20 positive leukemia cells[47,48]. The dependence of SynNotch CAR activity on the expression of two tumor antigens could potentially increase the risk of tumor escape due to loss or attenuation of either antigen.

In summary, we developed a SynNotch-based GD2-B7H3 CAR-T cell and demonstrated its specificity and efficacy against NBL. We observed superior metabolic fitness and lower exhaustion profile of gated CAR-T cells that may provide additional advantages over conventional CAR-T cells. Furthermore, the efficacy and safety profile of our GD2-B7H3 T-cell construct in solid tumor models should encourage both preclinical and clinical application of this technology for the treatment of solid tumors.

## Methods

**Cell lines**. NB9464D is derived from murine TH-MYCN mice backcrossed to C57/B6J and transduced with murine GD2-GD3 synthetase. 282, 844 murine NBL cell lines were derived from TH-MYCN 129/SvJ mice. CHLA136, CHLA255, SK-N-(BE)2, LAN5, LAN6 human NBL cell lines were derived from patients with progressive disease. Human cell lines were either established at CHLA or obtained from the Children's Oncology Group (COG) Cell Culture and Xenograft Repository (www.COGcell.org). CHLA255 cells have a high level of GD2 expression and express the c-MYC protein, thereby representing high-risk, undifferentiated NBL lacking MYCN proto-oncogene amplification. CHLA136 cells have a high level of GD2 expression and have genomic amplification of MYCN. GD2 expressing LAN6 (LAN6^GD2+) cell lines were constructed by transducing wild-type LAN6 cells with GD2 synthase (B4GALNT1) and GD3 synthase (ST8SIA1). GFP and Luciferase-positive LAN6^GD2+ cells were subsequently generated for in vitro and in vivo experiments. All cell lines were tested for the presence of mycoplasma contamination (MycoAlert™ Mycoplasma Detection Kit, Lonza) and authenticated using Short Tandem Repeat (STR). For some experiments, cell lines were transduced with luciferase (firefly) or eGFP and then sorted to obtain a > 99% positive

population. Cell lines were maintained in culture with IMDM (Gibco) supplemented with 10% fetal bovine serum and 100 UI/ml penicillin/streptomycin (Gibco). For all functional studies, primary cells were thawed at least 12 h before experiments and rested at 37 °C.

**Immunohistochemistry**. Immunohistochemistry (IHC) of formalin-fixed paraffin-embedded (FFPE) tissue was performed using antibodies against mouse CD3 (Thermofisher), human CD3 (Thermofisher), and PHOX2B (Thermofisher) at 1:150 dilution. Dual IHC staining was performed sequentially on a Leica Bond-IIITM instrument using the Bond Polymer Refine Detection System (Leica Microsystems). Heat-induced epitope retrieval was performed for 20 min with ER2 solution (Leica Microsystems AR9640). Incubation time with the CD3 antibody was 15 min, followed by 8 min post-primary step and 8-min incubation with polymer HRP, and then endogenous peroxidase was blocked for 5 min followed by 10 min in DAB. After the first antibody staining was completed, slides were incubated with anti-PHOX2B antibody for 15 min, followed by post-primary AP for 20 min and post polymer for 20 min. Subsequently, slides were stained with Fast red for 7 min. Slides were washed three times between each step with a bond wash buffer or water. All experiments were done at room temperature. IHC stain was similarly performed with murine CD3.

**Generation of murine CAR constructs and CAR-T cells**. To construct the murine GD2-28z CAR, 14G2a GD2 ScFv sequence was cloned into previously constructed mouse CD28-CD3z CAR in MSCV retroviral backbone using standard in-fusion cloning. Mouse 4-1BB intracellular signaling domain (UniProtKB - P20334), CD8 transmembrane, and the hinge were synthesized by Integrated DNA Technologies (IDT). CD28 was replaced with CD8 hinge/transmembrane and 4-1BB to construct GD2-4-1BB-CD3Z second-generation mouse CAR. To improve CAR expression, the linker joining heavy and light chain scFv was modified to (Ser (GlyX4)) X4. Eco packaging cell lines were transfected with retroviral transfer plasmids using lipofectamine 2000 using the standard protocol. Supernatants were harvested 24 and 48 h after transfection and concentrated by ultracentrifugation 4 h at 18,500 × g. Concentrated viral particles were used to transduce T cells 24–36 h after stimulation, pending adequate activation, monitored by cell size. Viral particles in PBS were added to retronectin coated wells (25 mg/ml) at five to ten infectious particles per T cell and centrifuged for 90 min at 2500 × g to increase binding of retronectin to viral capsids. Stimulated T cells were immediately added to virus-bound wells and spinoculated at 800 × g for 30 min at 31 °C. Cells were kept in culture for 48 h before analyzing CAR expression.

**Generation of human CAR constructs and CAR-T cells**. The second-generation anti-B7H3 chimeric antigen receptor (B7H3 CAR) features an anti-B7H3 scFv (BRCA84D (MG27A), MacroGenics Inc., US patent # 8802091 B2), CD8 hinge (UniProtKB—P01732), 4-1BB costimulatory domain (UniProtKB—Q07011) and CD3-ζ (UniProtKB—P20963) signaling domain (Supplementary Fig. 1). Production of CAR-expressing T cells was performed, as previously described[39,49,50]. In brief, lentiviral supernatants were produced via transient transfection of the 293T cell line and concentrated with PEG-8000, as previously described. Human T cells were isolated from peripheral blood mononuclear cells (PBMC) obtained from healthy donors (EasySep™ Human T Cell Isolation Kit). CAR-T cells were produced from PBMC of at least two unique healthy donors for all experiments. T cells were activated with anti-CD3/CD28 beads (Life Technologies) in a 3:1 bead:cell ratio with 50 IU/mL IL2 for 24 h. Anti-CD3/CD28-activated T cells were transduced by placing cells on plates pre-coated with Retronectin (20 ng/ml, Takara) and concentrated lentivirus. Media and IL2 were changed every two days until harvesting cells 10–14 days post transduction. Transduction efficiencies were routinely 60–70% for all constructs. T cells co-expressing SynNotch receptors and responding genes were generated by transducing a 50:50 mixture of both viruses simultaneously. B7H3-BBz CAR-T and GD2-B7H3-tCD19 cells were enriched using anti-Fab (murine) and anti-CD19 PE-conjugated antibodies followed by anti-PE nanobead magnetic separation (MojoSort™ Human anti-PE Nanobeads, Biolegend) per the manufacturer's recommendation.

SynNotch receptors were constructed by linking signal peptide derived from the human CD8 to an anti-GD2 or anti-B7H3 single-chain variable fragment (scFv). All sequence was reverse translated, codon-optimized, and synthesized (by Integrated DNA Technologies, San Diego, CA). The resulting product was subcloned into pHR_PGK_antiCD19_SynNotch_Gal4VP64 (addgene Plasmid #79125), replacing the anti-CD19 region. The CAR construct for the gated system was generated by subcloning GD2-BBz or B7H3-BBz in pHR_Gal4UAS_IRES_mC_pGK_tBFP (addgene plasmid #79123) upstream of IRES. Subsequently, the IRES_mC_pGK_tBFP was removed to generate tag-free UAS-B7H3-BBz. The extracellular portion of the human CD19 gene (UniProtKB—P15391) with pGK promoter was added to the UAS-B7H3-BBz to generate UAS-B7H3-BBz-tCD19 that was subsequently used for enrichment. pHR_Gal4UAS_tBFP_PGK_mCherry (addgene plasmid# 79130) was used for reporter BFP studies. A CD123-B7H3 CAR with tCD19 system was also developed as an experimental control since CD19-B7H3 could lead to self-engagement and stimulation of T cells through recognition of truncated CD19.

### CD107a degranulation, cytokine analysis, and cytotoxicity assay

*CD107a degranulation.* CD107a degranulation assays were conducted by co-culturing T cells with neuroblastoma tumor cells for 4 h in the presence of 2 μM monensin, followed by staining with CD107a antibody (Biolegend) and evaluation by flow cytometry. The degranulation assay for the gated CAR-T cells was performed similarly, except the cells were initially co-cultured for 24 h with target cells to activate CAR expression, and then subsequently, the 4-h degranulation assay was conducted. *Cytokine analysis:* Cytokine production was assayed by ELISA of supernatant harvested from wells containing CAR-T cells co-incubated with target tumor cells at a 1:1 ratio ($1 \times 10^6$ cells each) for 48 h. Harvested supernatants were analyzed using Human Cytokine Array Pro-Inflammatory Focused 13-plex (HDF13) (Eve technologies, Calgary, AB Canada). *Cytotoxicity assay:* Calculated T-cell effector to tumor target (E:T) ratios were based on the transduction efficiency of CAR-T cells. The total number of T cells in cytotoxicity experiments was adjusted to remain the same across experimental groups by the addition of untransduced T cells. GFP-labeled tumor cells were seeded in 96-well plates ($2.5 \times 10^3$ cells/well) followed by the addition of CAR-T cells in defined E:T ratios ($n = 4$ per E:T evaluation) in a final well volume of 200 μl of T-cell media and placed in an IncuCyte® S3 Live-Cell Analysis System (v2018B, Essen Bioscience). The integrated green fluorescent intensity or green fluorescent-positive cell count per well was calculated using IncuCyte software, standardized to baseline wells.

### Flow cytometry and cell enrichment

All samples were analyzed using an LSR II or FACSAria II (FACSDiva software (v8.0.), BD Bioscience), and data were analyzed using FlowJo X 10.6. GD2 and B7H3 SynNotch receptors, GD2, and B7H3 CAR were detected using goat anti-mouse F(ab')$_2$ fragment-specific antibody (Jackson ImmunoResearch) and using B7H3 protein (Sino Biologicals) conjugated to PE fluorochrome (Abcam). Expression of upstream activation sequence (UAS) B7H3-PgK-tCD19 was detected using an anti-CD19 antibody (Biolegend). For cell number quantitation, CountBright™ beads (Invitrogen) were used according to the manufacturer's instructions. In all analyses, the population of interest was gated based on forward, and side scatter characteristics followed by singlet gating. Live cells were gated using Live Dead Fixable Zombie UV (Biolegend). Antigen density for GD2 and B7H3 was measured using QuantiBrite™ beads (BD Biosciences) and 1:1 PE-conjugated GD2 and B7H3 antibody (BD Biosciences) per manufacturer's recommendation. Intracellular staining was performed using Biolegend reagents per the manufacturer's protocol (https://www.biolegend.com/en-us/protocols/intracellular-flow-cytometry-staining-protocol). In summary, CAR-T and target cells were co-cultured for 24 h before cytokine analysis with the addition of monensin and brefeldin A 6 h before harvesting cells. Harvested cells were stained with Live/Dead marker, anti-CD3, and anti-Fab antibodies, followed by fixation and permeabilization. Then fixed cells were stained for individual intracellular cytokines separately.

### Analysis of metabolic parameters

Mitochondrial function was assessed with an extracellular flux analyzer (Seahorse Bioscience). Individual wells of an XF24 were coated with CellTak per the manufacturer's instructions. The matrix was adsorbed overnight at 37 °C, aspirated, air-dried, and stored at 4 °C until use. Mitochondrial function was assessed following a 48-h co-culture of T cells with tumor cells at a 1:1 ratio. T cells were subsequently isolated using CD45 nanobeads (Biolegend). Cells were suspended in XF assay medium (non-buffered RPMI 1640) containing 5.5 mM glucose, 2 mM L-glutamine, and 1 mM sodium pyruvate and seeded at $1 \times 10^6$ cells per well. The microplate was centrifuged at $1000 \times g$ for 5 min and incubated in standard culture conditions for 60 min. During instrument calibration (30 min), cells were switched to a $CO_2$-free (37 °C) incubator. XF24 assay cartridges were calibrated in accordance with the manufacturer's instructions. Cellular OCRs were measured under basal conditions following treatment with 1.5 μM oligomycin, 1.5 μM FCCP, and 40 nM rotenone with 1 μM antimycin A (XF Cell Mito Stress kit, Seahorse Bioscience).

### Gene expression and GSEA analysis

GD2-B7H3 T cells enriched for tCD19, B7H3 CAR-T cells, and UT cells were co-cultured with CHLA255 NBL cells at 1:1 E:T ratio. T cells were then isolated with anti-CD45 nanobeads (Biolenged) to prevent contamination by tumor cells. RNA was extracted from cells using the RNeasy Mini Plus Kit (Qiagen). RNA quality was verified, and NGS was performed by Novogene (Sacramento, CA). Genes were selected at ≥twofold difference and $P$ value = <0.05. Heatmaps were generated with expression data normalized to mean of zero and standard deviation of one. GSEA analysis was performed using default parameter settings using published gene sets (MSigDB, https://www.gsea-msigdb.org/gsea/ msigdb). Significance was defined as FDR < 0.05. R (ver 3.4) was used to visualize the data.

### Statistics

All statistics were performed as indicated using GraphPad Prism version 8.0 (La Jolla, CA). Two-tailed Student's $t$ test was used to compare two groups; in an analysis where multiple groups were compared, one-way analysis of variance (ANOVA) was performed with Holm–Sidak correction for multiple comparisons. When multiple groups at multiple timepoints/ratios were compared, Student's $t$ test or ANOVA for each time, points/ratios were used. Each graph represents three biological replicates unless otherwise noted in the figure legend. The $P$ value of each experiment is inserted in the graph when applicable, and "ns" means "not significant" ($P > 0.05$).

### In vivo studies

For orthotopic murine studies, the capsules of the left kidneys of C57BL/6J mice (6–8 weeks old) were inoculated with NB9464D$^{GD2+Luc+}$ murine NBL cells ($1 \times 10^6$ cells/mouse) under anesthesia and using approved surgical techniques. Tumors were allowed to grow for 34 days (100% take rate) to generate syngeneic NBL models. Daily intraperitoneal (i.p.) injection of cyclophosphamide (110 mg/kg/day) and topotecan (0.4 mg/kg/day) was started on day 35 and continued for 5 days as a lymphodepletion strategy in immunocompetent mice to allow for T-cell expansion and to mimic human CAR-T-cell protocols[31,51]. The choice of chemotherapy for lymphodepletion reflects that used in treating patients with high-risk neuroblastoma. Similarly, NB9464D$^{GD2+Luc+}$ murine NBL cells ($1 \times 10^6$ cells/mouse) were injected intravenously (i.v.) in NSG mice to develop metastatic murine NBL followed by chemotherapy (described above) before CAR-T infusion. Xenograft studies were performed using NSG mice (NOD.Cg-Prkdc$^{scid}$ ILrg$^{tm1Wjl}$/SzJ, Jackson Laboratory) at 6–12 weeks of age. Equivalent numbers of male and female mice were used in each experiment. The NSG mice were inoculated intravenously (i.v.) with CHLA255, LAN6, LAN6$^{GD2+}$, or CHLA136 luciferase-expressing NBL cells ($1 \times 10^6$ cells/mouse). Mice were injected with $1 \times 10^6$–$1 \times 10^7$ transduced or untransduced T cells depending on the experiment. Similar to in vitro experiments, the total number of T cells given to animals was the same across the treatment cohorts, and untransduced T cells were added to CAR-T cells based on their transduction efficiency. Xenogen IVIS Lumina (Caliper Life Sciences) was used to monitor disease burden and progression. Xenogen images of mice were taken 15 min after injection of 1.5 mg D-luciferin (Caliper Life Sciences) intraperitoneally. For all experiments, the xenogen exposure time was set at 3 min. Luminescence images were analyzed using Living Image software (Caliper Life Sciences). All raw bioluminescence data and corresponding summary graphs are presented in Supplementary Data, where the black arrow points to the timing of the CAR-T or UT T-cell injection. Peripheral blood was collected from the retro-orbital venous system or tail vein. All mice were kept under specific pathogen-free conditions and housed in the Saban Research Institutes' Animal Facility. All animal studies were conducted in accordance with and approved by the Children's Hospital Los Angeles Institutional Animal Care and Use Committee (IACUC). Animals injected with tumor cell lines were monitored weekly until developed clinical symptoms, which after was monitored daily. Animals meeting study criteria, including weight loss >15%, significant change in behavior, seizure, or decreased mobility, were euthanized according to the approved procedures. The primary endpoint of all animal studies was survival, which was analyzed using the long-rank Mantel–Cox test. All studies were unblinded, and no animals were excluded from the analysis.

### Single-cell dissociation of brain tissue

Hearts of anesthetized mice were perfused with ice-cold saline for a minimum of 10 min before the harvest of brain tissue. Single-cell suspensions were performed using a gentleMACS dissociator (Miltenyi). Briefly, each cerebellum was cut into half in symmetric pieces—one half of tissue used for IHC staining and the other half assigned for single-cell processing. Approximately 0.5 g of the cerebellum was digested in 3 ml RPMI with 150 μl collagenase (40 mg/ml PBS) and 100 μl dispase (32 mg/ml PBS) in a gentleMACS C tube. This was kept in a 37 °C water bath with mild shaking for 20 min. The cell suspension was filtered through a 70-μm cell strainer and centrifuged at $300 \times g$ for 10 min. The cell pellet was suspended in the required volume of MACS buffer. Counting beads were added to maintain equal rations between samples.

**Reporting summary.** Further information on research design is available in the Nature Research Reporting Summary linked to this article.

## Data availability

The data that support the findings of this study are available from the corresponding author upon reasonable request. Raw RNAseq data supporting the findings of this study have been deposited in the National Center for Biotechnology Information Gene Expression Omnibus (NCBI-GEO) under accession number GSE161942. Databases used for collecting gene and/or functional pathway information include the Molecular Signatures Database (https://www.gsea-msigdb.org/gsea/msigdb). Source data are provided with this paper.

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

## Acknowledgements

This work was supported in part by P01-CA217959 (S.A., PI R.C. Seeger, J. Maris) from the National Cancer Institute, United States (S.A.), CA170257P1 from the Department of Defense (S.A. and M.D.H.), T.J. Martell Foundation (S.A.), and Nautica Malibu Triathlon (S.A.). Soccer for Hope Foundation (S.A.), V Foundation (S.A.), Norris Foundation (S.A.), Hyundai Hope On Wheels (B.M.), Pediatric Physician-Scientist Training Program at Children's Hospital Los Angeles (B.M.), and an NIH T32 CA009659 training grant (R.T.). The Pathology Research Core Facility provided histological services in the Department of Pathology at Children's Hospital Los Angeles. The Saban Research Institute supports the small animal imaging core and flow cytometry core facilities at Children's Hospital Los Angeles. Thanks goes to Dr. Paul Sondel, Dr Malcolm Brenner, and Dr. Kucikichfer for kindly providing NB9464D, 14G2a GD2 ScFv sequence, and mouse CD28-CD3z CAR, respectively.

## Author contributions

B.M. and S.A. conceived the study, designed experiments, analyzed the data, and wrote the paper with input from all authors. B.M and S.M. performed experiments, analyzed the data, and interpreted the results. S.J., R.T., and L.H. assisted with experiments and injected tumors. H.B., M.D.H., and D.M.B. provided material support and design of murine experiments. H.S. interpreted pathology slides and contributed to discussions.

## Competing interests

The authors declare no competing interests.
