## [Peer Review File · Nature Communications]

Reviewers' comments:

Reviewer #1 (Remarks to the Author):

Moghimi, et al engineer novel GD2⁺B7H3⁺ synNotch -> CAR T cells that preferentially kill CHLA255 GD2⁺/B7H3⁺ cell lines compared to GD2⁻/B7H3⁺ LAN6 cell lines in vitro. They propose that this novel circuit can reduce ON target, OFF tumor toxicity in neuroblastoma. The manuscript also demonstrates that (i) a CD28 CAR is more efficacious than 41BB in clearing tumors, but also exhibits lethal neurotoxicity and (ii) the GD2⁺B7H3⁺ synNotch -> CAR T cells have clear metabolic and exhaustion benefits compared to constitutively expressed B7H3 CAR T cells. If the following major comments are addressed, then this manuscript would represent a significant contribution to the field of CAR T cell engineering by (i) demonstrating a second set of tumor-associated antigens capable of enhancing CAR T cell efficacy and safety (after Srivastava, 2019) and (ii) demonstrating for the first time that gating CAR expression via synNotch transcriptional control reduces exhaustion and metabolic markers in synNotch T cells.

Major comments:

1. The paper would be strengthened if the choice of gating and killing ligands were explained in more detail in the Introduction. In its current form, it appears that neuroblastoma has two tumor-associated antigens (TAAs): B7H3 and GD2. Gating on either GD2 or B7H3 should increase the specificity of the therapy, but it is curious that the GD2 antigen was chosen as the gating antigen given that the main ON target, OFF tumor side effects of GD2-related therapies are pain and motor neuropathies (since GD2 is also expressed in the cerebellum and peripheral nerves). B7H3 is a TAA that is aberrantly expressed in neuroblastoma; its expression in healthy cells is limited to non-neuronal tissues (liver, prostate, adrenal gland, & stimulated monocytes). Would B7H3 therefore be a better gating antigen, given both the limited tissue expression of B7H3 and the fact that "the majority of failures of animals treated with multiple infusions occurred in brain and bone marrow in contrast to those treated with a single dose of GD2⁺B7H3⁺" and that GD2 is expressed in the brain?

2. Moghimi, et al compare CHLA255 GD2⁺/B7H3⁺ and GD2⁻/B7H3⁺ LAN6 cell lines in vitro to demonstrate the enhanced antigen specificity of their GD2⁺B7H3⁺ circuit, but their in vivo experiments do not have the controls necessary to support this conclusion (Fig. 2J; Fig. 3H). In order to draw the conclusion that this circuit is highly specific, a comparison between mice bearing GD2⁺/B7H3⁺ and GD2⁻/B7H3⁺ tumors is needed in analogy to literature precedent (Roybal 2016; Srivastava, 2019). This could be accomplished by: (i) comparing CHLA255 and LAN6 tumors, (ii) comparing CHLA255 and CHLA255-GD2 KO tumors, (iii) comparing LAN6 and LAN6-GD2 KI tumors. It is possible that differences in tumor engraftment make comparisons between CHLA255 and LAN6 tumors impractical, but further engineering of one or both of the cell lines is feasible.

Minor comments:

1. Figure 1:

- a. Can they include experiments at 1:1 E:T?
- b. The degranulation assay could be omitted given that the killing data are quite clear.

2. Figure 2:

- a. Panel "I": Can they include killing curves for the LAN6 cells (GD2⁻ / B7H3⁺), as in Fig. 2F (degranulation)?
- b. Related to Major Comment #2: Where are the data comparing CHLA255 vs. LAN6? This is what determines specificity. If these data are not included, then the title should perhaps be adjusted to "in vitro specificity".

3. Fig. 3:

a. What motivated engineering the cells to enrich the circuit? This is interesting, and if emphasized could be a more significant result.

b. Related to Major Comment #2: This figure makes the relevant comparison – CHLA255 vs LAN6 – in the in vitro assays, but does not make this comparison in the in vivo assays. Although the in vitro data indicate that the GD2^{B7H3} synNotch -> CAR circuit would be selective in vivo, the standard in the literature is to confirm this selectivity in vivo (Roybal 2016; Srivastava, 2019).

c. Repeat dosing of GD2^{B7H3} T cells in NSG mice bearing CHLA255 tumors (Fig. 3) results in the following: “the majority of failures of animals treated with multiple infusions occurred in brain and bone marrow in contrast to those treated with a single dose of GD2^{B7H3}” – this is a curious result given that the gate is intended to reduce neurotoxicity. Can the authors speculate as to why this would be the case? Is the issue systemic inflammation, or does repeated dosing lead to ON target, OFF tissue defects?

4. Under what conditions would the current gating strategy break down? For example, the gating strategy in Srivastava, 2019 broke down when the normal and cancerous tumors were mixed. Would a similar effect take place with GD2 and B7H3?

5. Stylistic comment #1: it would be helpful to the reader if the title addressed the principle conclusions of the study – namely that a SynNotch circuit can reduce ON target, OFF tumor toxicity in a metastatic neuroblastoma model.

6. Stylistic comment #2: Overall the figures are accurately and helpfully labeled and include appropriate error bars, but the figure labels are too small to read across the board.

Reviewer #2 (Remarks to the Author):

The manuscript by Moghimi et al. presents a novel application of the recently introduced SynNotch module for tailoring anti-tumor CAR-T cell responses. While CAR-T cells (CARTs) targeting CD19 are in clinical use for B cell derived malignancies, their long-term efficiency against solid tumors is limited due to exhaustion of the transferred T cells and subsequent lack of persistence or by side effects including cytokine-release syndromes and neuropathies (referred to as ICANS, immune cell-associated neurologic syndrome). These limitations might be overcome by the recently introduced synthetic notch module (SynNotch) allowing for coupling of two different tumor-associated antigens (TAAs) to increase selectivity of the CAR-T response while possibly reducing side effects. The authors used this approach to link the TAAs GD2 and B7-H3 via SynNotch. The resulting CARTs were analyzed in the context of neuroblastoma, a solid tumor of childhood, which usually expressed both GD2 and B7-H3.

General remarks:

Work in this manuscript appears to be performed technically sound. Results are presented in a way that enables other researchers to reproduce the work.

That said, I had a hard time following the logic in the current version of the manuscript, which first tells the reader that B7-H3 directed CARTs cure metastatic neuroblastoma, which is in line with results published earlier this year on other solid tumors including childhood malignancies. Second, we learn that re-engineering of CARTs targeting GD2 did not work and only then we are informed on the results of linking GD2 and B7-H3 via SynNotch. I consider the following order as easier to follow: 1.

Engineering GD2-CAR T cells by introducing CD28 or 4-1BB co-stimulatory molecules does not work

due to toxicity issues and lack of efficacy, respectively; 2. As GD2-CAR T cells have been proven efficacious on NB models and given the encouraging results of others using B7-H3 CARTs, the authors compared the GD2-SynNotch-B7-H3 CAR-T cells to CAR-T cells expressing B7-H3 only.

Major points:

-GD2-SynNotch-B7-H3 CAR-T cells yield favorable biomarker and metabolic profiles, while delaying tumor clearance compared to CARTs targeting B7-H3 only. If the authors want to argue for their SynNotch construct on the ground of a favorable molecular profile, they need to add data showing a benefit over the "B7-H3 only" constructs, e.g. by re-challenging the mice with tumor cells upon tumor clearance after 4 weeks. Emergence of exhaustion markers in the "B7-H3 only" CARTs could otherwise be interpreted as a sign of lack of target cells. Therefore, a re-challenge could inform on long-term tumor control conferred by either of the CARTs used here. Additionally, it would also be interesting, if the assumed exhausted phenotype of B7-H3 only CAR-T cells would be reversed by simultaneous treatment with antibodies directed against the identified marker proteins, e.g. LAG-3 and PD-1.

-Higher risk for development of ICANS have been suggested to be linked to CD28-containing constructs, which has been reported in at least two major studies (Neelapu et al., NEJM, 2017 and Koechenderfer et al, JCO, 2015). Additionally, the observation that rapid expansion of CARTs containing CD28 has been linked to breakdown of the blood-brain-barrier leading to fatal outcomes in the JCAR015 trial could have come as a warning sign. The authors are strongly encouraged to rephrase the entire paragraph on p9 accordingly and also to critically discuss these findings. Nevertheless, I consider the results presented in figure S3 as highly important as they demonstrate a link between (rapid expansion of) CD28-containing CARs and ICANS in a solid tumor model. Importantly, the authors need to describe the immune cell infiltrates by additional staining for macrophages and activated microglia (e.g. CD68). It would also be interesting to know, if the CD3-positive cells found in the brain were CAR-T cells.

-The authors stress that targets B7-H3/CD276 and GD2 are expressed independently of the MYCN oncogene. As MYC proteins confer an immunosuppressive environment, did the authors observe any differences in cytokine profiles in tumors with high levels of MYCN vs low levels of MYCN? In the metastatic model presented here, intravenous injection of tumor cells was used, which does not fully reflect the microenvironment of a growing tumor. It would be interesting if MYC-related differences were to be observed in s.c. transplant model or, if available, in one of the available MYC-transgenics for neuroblastoma.

Fig. S6: This is an important experiment regarding leakiness of the B7H3 CAR. Unfortunately, data interpretation is hampered by incomplete labelling (no description of the x-and y-axis in A) and description of the experimental read-out. Please give scores of Phox2B and CD3 positive cells to evaluate if there is a significant difference in tumor cell number and T cell infiltration depending on E:T ratios of UT and the CD123/B7H3 CAR.

Minor: several typos need to be addressed, e.g. p 33 (Biolenged = Biolegend), legend to Fig. S6 (in = is).

Reviewer #3 (Remarks to the Author):

The authors describe the pre-clinical development of a CAR T-cell therapy approach for childhood solid tumour neuroblastoma. As surface antigens that have a truly tumour specific expression pattern and hence provide optimal targets for a CART approach are limited, they focus on developing a CAR incorporating an SynNotch gate - which is effectively an 'AND' gate - limiting CAR T-cell activation to

the presence of two tumour associated antigens in this case GD2 and B7-H3. First, they demonstrate efficacy - in the absence of toxicity - of B7-H3 directed CAR T-cells in an NSG model of metastatic neuroblastoma. Then they describe, in an orthotopic syngeneic neuroblastoma model, that mice treated with CAR T-cells directed to GD2 using a 14g2a-based CAR develop toxicity which they interpret as on-target off-tumour toxicity. They then use a SynNotch approach, referred to as GD2[^]B7-H3 CAR, to create an AND gate with expression of the B7-H3 CAR gated by binding to an affinity enhanced version of the GD2 scFv. Efficacy of GD2[^]B7H3 CAR T-cells is demonstrated in a metastatic model of neuroblastoma although tumour control is reduced as compared to standard B7-H3 CAR T-cells. GD2[^]B7-H3 CAR T-cells were shown to have a reduced oxygen consumption rate and reduced expression of immune inhibitory receptors/transcription factors associated with T-cell exhaustion as compared to B7-H3 CART cells. This metabolic/phenotypic profile of GD2[^]B7H3 may translate into improved in vivo CAR T-cell persistence.

Major comments:

The crux of this paper is the development of an AND gate to minimize risk of on-target off-tumour toxicity of CAR T-cells directed against an tumour associated antigen which is also expressed on some normal tissues. The SynNotch approach to create an AND gate is not novel and has been described by Srivastava et al, *Cancer Cell* 2019 [PMID:30889382] to prevent on-target off-tumour toxicity albeit in the context of a different tumour type/tumour associated antigens.

The authors start out with describing CARs targeting two single antigens expressed on neuroblastoma B7-H3 and GD2. They describe the efficacy of a B7-H3 specific CAR in NSG models of neuroblastoma. Again this is not novel as Du et al [*Cancer Cell* 2019, PMID: 30753824] have described efficiency of B7-H3 CAR T-cells in the same CHLA255 model of neuroblastoma albeit using a CAR based on a different B7-H3 specific antibody. Similar to described findings by Du et al the authors do not see any treatment related toxicity in this model.

The authors then report an unexpected finding using a GD2-directed CAR based on GD2-specific antibody 14g2a. Here a different model of neuroblastoma is used i.e. a syngenic model with renal capsule engraftment of a murine neuroblastoma cell line transduced to express GD2 where mice are treated with chemotherapy prior to CAR T-cell administration. They report that mice treated with T-cells transduced with a GD2-CAR incorporating CD28zeta endodomains developed signs of neurotoxicity whereas toxicity was not observed in mice treated with T-cell transduced with GD2-CAR incorporating 41BBzeta endodomains. The authors interpret signs of toxicity as on-target off-tumour toxicity however, the cause of the observed toxicity is not sufficiently explored to draw this conclusion. Signs of neurotoxicity can be caused through different mechanisms including (1) neurotoxicity secondary to CART activation/inflammation as described in patients treated with CD19-specific CAR T-cells, (2) local inflammation due to homing of CAR T-cell to tumour deposits in the brain or (3) on-target off-tumour toxicity due to GD2 expression on normal brain tissues. The data shown to explore the cause of the observed toxicity is limited to immunohistochemistry of cerebellum/spine of one mouse in the GD2-CAR_CD28zeta endodomain group (Supplementary Figure 3). This showed presence of CD3+ T-cells - however no comparison of the same staining for mice from the other treatment groups is provided. No specific staining for CAR expressing T-cells is performed. It is not established if T-cells are infiltrating in GD2-expressing areas of the CNS. A second query about the result of this experiment is that the tumour burden at baseline does not appear equal between treatment groups. A graph showing quantified BLI signal over time should be included in addition to individual BLI images.

The authors then go on to generate a syn-Notch AND gate with B7-H3. They do not demonstrate on-target off-tumour toxicity with B7-H3 and Du et al (their ref 26) shows no on-target off-tumour toxicity with B7-H3 in an immunocompetent mouse model. Given clinical experience with (plain)

14g2a and no published data showing on-target off-tumour toxicity, there is relatively little justification of making a GD2/B7-H3, especially since the potency of the gated CAR is lower.

In summary, the main questions are:

- Why was a different model used to assess the GD2-CAR?
- How can the authors demonstrate that the toxicity seen in this model is due to on-target off-tumour toxicity.
- As there is existing clinical data (Louis et al, Blood 2011 [PMID:21984804] targeting GD2 in neuroblastoma where neurotoxicity was not reported, how relevant as the described findings in this mouse model?
- The generation of a B7-H3/GD2 AND gate CAR is not justified - this would be a logical approach if expressing non-cross reactive on-target toxicity between the two target antigens - otherwise restricting B7-H3 recognition to GD2 expressing tissue risk increasing antigen escape

Minor comments:

Figure 1b: left - how are CAR T-cells detected, right; please include control of CAR T-cells co-cultured with a non-B7-H3 expressing cell line

Figure 1c-f: while cytokine secretion, T-cell proliferation and cytotoxicity of B7-H3 CART cells is demonstrated as compared to non-transduced T-cells, no non-B7-H3 expressing target cells are included to demonstrate B7-H3 specificity of CART activity.

Figure 2: G-I shown is T-cell proliferation, cytokine secretion and cytotoxicity of GD2+/B7-H3+ CHA255, but not absence of CART activity against a GD2-/B7-H3+ cell lines such as LAN-6.

Figure 3J: the colours use for mice treated with a single or repeated injections of GD2⁺B7-H3 CART is similar and hence this figure is hard to read.

Figure S3: how were murine GD2-specific CARs constructed?

Discussion: 'use of the SynNotch gated system was not associated with neurotoxicity even though the SynNotch receptor contains the high affinity GD2(E110K scFv) - the SynNotch receptor was tested in the CHLA NSG model whereas toxicity was observed in the syngeneic NB9464D(GD2+luc+) model. Moreover, to test this as outlined above, a B7-H3⁺GD2(E110K) should be used.

Reviewer #4 (Remarks to the Author):

In this study Moghimi B. et al. report that a SynNotch gated CAR-T cell targeting GD2 as the gate and B7H3 as the target of the CAR controls Neuroblastoma tumor growth in a xenograft mouse model of Neuroblastoma. Moreover, in vitro GD2⁺B7H3 CAR-T cells co-cultured with NBL cells are more resistant to exhaustion and show greater metabolic fitness, compared to 'conventional' B7H3 CAR-T cells.

In my opinion, the manuscript is well written, easy to follow and present original and interesting data. The SynNotch approach is an interesting novel approach that will be of interest. It gives extra specificity to CAR-T cells. GD2⁺B7H3 CAR-T cells show in vitro high NBL killing efficacy, although the in vivo therapeutic efficacy shown is not so striking, since 'conventional' B7H3 CAR-T cells can reach the same potency. Also, the therapeutic efficacy of CD19 enriched GD2⁺B7H3 CAR-T is higher when therapy is started earlier (7 days post i.v. challenge with tumor cells), although still not as efficient as 'conventional' B7H3 CAR-T therapy. Efficacy of CD19 enriched GD2⁺B7H3 CAR-T is improved when three doses starting at day 14 after i.v. tumor cell injection are administered, demonstrating tumor growth control, prolonged mice survival, although all mice finally die.

It seems the most effective therapeutic CAR-T in vivo approach is still that with 'conventional' B7H3 CAR-T, in spite of higher expression of exhaustion markers and inferior metabolic fitness compared to GD2⁺B7H3 CART.

No data are presented on how long GD2⁺B7H3 CAR-T cells or any CAR-T cells can survive in mice and thus can be detected in blood. Can CAR-T cells be recovered in vivo?

It would be interesting to see also the Kaplan-Meier curve associated with the bioluminescence images in all figures.

My major concern resides on the issue of toxicity: since the GD2 associated neurotoxicity is detected in the syngeneic model, the final demonstration that SynNotch modification help to avoid toxicity needs to be demonstrated in the same syngeneic immunocompetent NBL model. In ref 26 and in the discussion the authors properly cite the in vivo use of a B7H3 CAR-T approach in an immunocompetent mouse model, so it seems feasible to use also GD2⁺B7H3 CAR-T. In addition, in the syngeneic model also CAR-T associated toxicity can be studied (circulating cytokines).

Moreover, the author state that CD28 costimulatory domain is associated with neurotoxicity (figure S3 and discussion) as mice receiving GD2-28z CAR-T showed deep toxicity, while mice receiving GD2-BB CAR-T did not. How can the CD28 costimulatory domain be responsible for neurotoxicity?

Just a comment: An additional improvement of SynNotch CAR-T cells may be the insertion of a safety switch that can induce CAR-T cell apoptosis to control their in vivo expansion and prevent or limit any adverse reaction, especially in view of a future translation to the clinic.

Specific comments:

1) Have the in vivo experiments been performed more than once?

2) In the Introduction section, at the bottom of the first page the authors mention Fig. 2A to explain the SynNotch gating strategy. It sounds strange to me to claim in the introduction "Fig. 2", while Fig 1 has not been mentioned yet.

3) In the Results section, in the paragraph entitled: "Construction of highly specific and effective GD2⁺B7H3 CAR-T cells": since you show GD2-28z CAR-T neurotoxicity which is the basis for your future choice of the SynNotch approach, this should be a figure, not a supplementary Figure, see point 4; in Figure S3, in panel A description and in Materials and Methods the therapeutic regimen given to mice before administration of GD2-28z, GD2-BBz CAR-T cells or UT is described. Why did the authors decide to treat syngeneic mice with cyclophosphamide and topotecan before CAR-T administration? No mention or explanation in the Result section about the chemotherapeutic regimen is given, and it should.

4) In the discussion section:

- "...herein we demonstrate the novel finding that neurotoxicity" Sounds a bit strange since the figure that reports neurotoxicity in mice receiving GD2 CAR-T cell therapy is a supplementary figure. This should be a regular figure. In the same sentence you imply that CD28 costimulatory domain in GD2-CAR-T is associated with neurotoxicity.

Have you repeated the in vivo experiment more than once with a different GD2-28z CAR-T preparation?

It is not clear how you demonstrate that CD28 costimulatory domain may be responsible for neurotoxicity. If GD2 and CD28 associated toxicity is the reason claimed in the manuscript to be reduced/eliminated, you might not need to use a SynNotch gated strategy, but only modify (as you did) the costimulatory domain from CD28 to 4.1BB. In fact, the authors report that mice treated with GD2-BBz CAR-T cells did not show neurotoxicity, but a limited therapeutic effect. The authors should better clarify this issue.

5) In the materials and methods section: no indications of how intracellular staining for cytokine production is performed, but in the figures (Fig. 1C, 3E) a contour plot is shown as representative. In general, are data represented in the histogram charts the mean values +/- SD?

Minor comments:

In the results section:

- second page "...100% overall survival over the 6 month observation period while the mice that received with UT cells or..." Is the word 'with' a misprint?
- A line below: "...Similar in vivo efficacy of B7H3 CAR-T cells was observed a second metastatic murine..." Is 'in' missing?
- Fig. 2B should be cited when explaining the GD2⁺BFP T cells.
- In "Enrichment of GD2⁺B7H3 T cells improves efficacy against NBL" can the authors explain what they mean with 'serving as a kill switch' in the sentence "...adding a truncated version of human CD19 (tCD19) to allow for enrichment and detection of T cells while also serving as a kill switch (Fig. 3A)."??
- The authors demonstrate the specificity of the GD2⁺B7H3 CAR-T SynNotch towards CHLA255 a GD2⁺B7H3⁺ NBL cell line and use as control LAN6 a GD2⁻B7H3⁺ NBL cell line. The control cell line should be shown not only for the cytotoxicity assay, but also for cytokine production.

Reviewer #1:

Moghimi, et al engineer novel GD2⁺B7H3⁺ synNotch -> CAR T cells that preferentially kill CHLA255 GD2⁺/B7H3⁺ cell lines compared to GD2⁻/B7H3⁺ LAN6 cell lines in vitro. They propose that this novel circuit can reduce ON target, OFF tumor toxicity in neuroblastoma. The manuscript also demonstrates that (i) a CD28 CAR is more efficacious than 41BB in clearing tumors, but also exhibits lethal neurotoxicity and (ii) the GD2⁺B7H3⁺ synNotch -> CAR T cells have clear metabolic and exhaustion benefits compared to constitutively expressed B7H3 CAR T cells. If the following major comments are addressed, then this manuscript would represent a significant contribution to the field of CAR T cell engineering by (i) demonstrating a second set of tumor-associated antigens capable of enhancing CAR T cell efficacy and safety (after Srivastava, 2019) and (ii) demonstrating for the first time that gating CAR expression via synNotch transcriptional control reduces exhaustion and metabolic markers in synNotch T cells.

Major comments:

1. The paper would be strengthened if the choice of gating and killing ligands were explained in more detail in the Introduction. In its current form, it appears that neuroblastoma has two tumor-associated antigens (TAAs): B7H3 and GD2. Gating on either GD2 or B7H3 should increase the specificity of the therapy, but it is curious that the GD2 antigen was chosen as the gating antigen given that the main ON target, OFF tumor side effects of GD2-related therapies are pain and motor neuropathies (since GD2 is also expressed in the cerebellum and peripheral nerves). B7H3 is a TAA that is aberrantly expressed in neuroblastoma; its expression in healthy cells is limited to non-neuronal tissues (liver, prostate, adrenal gland, & stimulated monocytes). Would B7H3 therefore be a better gating antigen, given both the limited tissue expression of B7H3 and the fact that “the majority of failures of animals treated with multiple infusions occurred in brain and bone marrow in contrast to those treated with a single dose of GD2⁺B7H3⁺” and that GD2 is expressed in the brain?

We appreciate the reviewer’s comment. We now explain the reasoning behind our initial choice (page 26). During the allotted time for revision, we constructed the B7H3-GD2 gated CARs (reverse of the original gating) and generated *in vitro* and *in vivo* data with these CARs (Figure 6 and discussion on Page 26). As we now state in the manuscript, the gated expression of CARs has been shown to have a decay trajectory after disengaging from tumor cells. Although GD2 has a more restricted pattern of expression compared to B7H3, its expression in CNS tissues could still invoke neurotoxicity from the circulating and decaying gated CAR T-cells. Our data shows that the reverse gating strategy (B7H3⁺GD2⁻ T cells) continues to be effective *in vitro* and *in vivo*. Although neurotoxicity was not observed using the reverse gated CARs, we did see CAR T-cell infiltration in the brain of mice with metastatic neuroblastoma treated while none was observed for the GD2⁺B7H3⁺ T cells. It is unclear if a greater number of gated CAR T cells undergoing decay (perhaps due to larger tumor) would cause neurotoxicity with these reverse gated CARs. These new findings emphasize the considerations that should be given about potential risk of toxicity to vital organs in selecting the CAR antigen in the SynNotch gated system, and in further research into the CAR decay mechanisms and kinetics of gated CAR systems.

2. Moghimi, et al compare CHLA255 GD2⁺/B7H3⁺ and GD2⁻/B7H3⁺ LAN6 cell lines in vitro to demonstrate the enhanced antigen specificity of their GD2⁺B7H3⁺ circuit, but their *in vivo* experiments do not have the controls necessary to support this conclusion (Fig. 2J; Fig. 3H). In order to draw the conclusion that this circuit is highly specific, a comparison between mice bearing GD2⁺/B7H3⁺ and GD2⁻/B7H3⁺ tumors is needed in analogy to literature precedent (Roybal 2016; Srivastava, 2019). This could

be accomplished by: (i) comparing CHLA255 and LAN6 tumors, (ii) comparing CHLA255 and CHLA255-GD2 KO tumors, (iii) comparing LAN6 and LAN KI tumors. It is possible that differences in tumor engraftment make comparisons between CHLA255 and LAN6 tumors impractical, but further engineering of one or both of the cell lines is feasible.

We agree with the reviewer that this is an important control. We were able to generate LAN6^{GD2+} KI cell lines through transduction with human GD2 and GD3 synthases and demonstrate the specificity and efficacy of the GD2⁺B7H3 gated CAR-T cells *in vitro* and *in vivo* metastatic mouse model (comparing LAN6 and LAN6^{GD2+} cell lines). These data are now part of Figure 4E and F.

Minor comments:

1. Figure 1:

a. Can they include experiments at 1:1 E:T?

Experimental data for E:T range from 0.5:1 to 10:1 are now included in the supplementary figure (Figure S4B).

b. The degranulation assay could be omitted given that the killing data are quite clear.

We feel that including degranulation data, at least for our first construct, demonstrates our workflow for validating the CARs. We have placed degranulation data for other CARs in supplemental Figures (Fig. S4). We have also placed the degranulation assay for murine CAR constructs into Supplemental Figure S2B.

2. Figure 2:

a. Panel "I": Can they include killing curves for the LAN6 cells (GD2- / B7H3+), as in Fig. 2F (degranulation)?

Cytotoxicity data of gated GD2⁺B7H3 CAR against LAN6 and LAN6^{GD2+} has been performed and placed in Figure 4E.

b. Related to Major Comment #2: Where are the data comparing CHLA255 vs. LAN6? This is what determines specificity. If these data are not included, then the title should perhaps be adjusted to "in vitro specificity".

We have now generated both *in vitro* and *in vivo* data. These data are presented in Figure 4. (also see the response to Major Comment #2).

3. Fig. 3:

a. What motivated engineering the cells to enrich the circuit? This is interesting, and if emphasized could be a more significant result.

SynNotch Gated T cells are generated using double virus transduction, one expressing synthetic notch receptor (gate vector) and the other includes the response elements and CAR construct (response vector). The transduction efficacy of the response vector cannot be functionally determined since the CAR itself is not expressed. Using the constitutive expression of truncated CD19 under its own promoter within the response vector allows us to detect the transduction efficiency of the response vector. This allowed us to enrich T cells using Anti-CD19 columns (available for clinical GMP use). We also chose

CD19 as it could be used as a target to eliminate CAR T-cells if needed (using anti-CD19 BiTEs). We have now expanded on this reasoning in Page 15, first paragraph.

b. Related to Major Comment #2: This Figure makes the relevant comparison – CHLA255 vs LAN6 – in the in vitro assays, but does not make this comparison in the in vivo assays. Although the in vitro data indicate that the GD2^{B7H3} synNotch -> CAR circuit would be selective in vivo, the standard in the literature is to confirm this selectivity in vivo (Roybal 2016; Srivastava, 2019).

We have now addressed this with the transduction of LAN6 to express GD2 and comparing LAN6 parental cells and GD2 expressing variant in both *in vitro* and *in vivo* setting. (see the response to Comment #2).

c. Repeat dosing of GD2^{B7H3} T cells in NSG mice bearing CHLA255 tumors (Fig. 3) results in the following: “the majority of failures of animals treated with multiple infusions occurred in brain and bone marrow in contrast to those treated with a single dose of GD2^{B7H3}” – this is a curious result given that the gate is intended to reduce neurotoxicity. Can the authors speculate as to why this would be the case? Is the issue systemic inflammation, or does repeated dosing lead to ON target, OFF tissue defects?

CHLA255 is an aggressive human NBL cell line that has the propensity to metastasize to the brain. The frequency of brain lesions in xenograft mice developed from CHLA255 is underestimated since tumor-bearing mice do not live long enough for CNS metastatic lesions to fully develop. The results show that infusions of gated CAR-T cells in higher burden disease control systemic disease but do not prevent tumors from escaping to the CNS. We believe this is likely due to a reduced rate in clearing tumors (lower cytotoxic efficiency) by the gated CAR cells which likely allow cells to escape into the CNS. In the future, more rigorous testing of gated CAR-T cells (through higher dosing and infusion regimens) may be able to suppress the disease burden fast enough to decrease these types of escapes. We now provide additional context to this finding on page 27, second paragraph.

4. Under what conditions would the current gating strategy break down? For example, the gating strategy in Srivastava, 2019 broke down when the normal and cancerous tumors were mixed. Would a similar effect take place with GD2 and B7H3?

We appreciate the reviewer’s valuable comment. The breakdown in this model is more likely to happen in high burden disease where hyperactivation of SynNotch gate can lead to T cell trafficking at sites positive for single antigen including some normal tissue. This might be considered an advantage for the SynNotch system where activation of a gated CAR by neighboring cells could exert anti-tumor activity against subpopulation that express gate antigen at lower levels. The temporal separation between gate activation and CAR-T signaling not only drives specificity but also could prevent antigen escape in tumors with a heterogeneous expression of the target antigens. We observed that in our reverse gating (B7h3^{GD2} CAR-T cells), the gated CAR trafficked to the brain without causing neurotoxicity. T cell trafficking to a vital organ such as brain regardless of their ability to exert cytotoxicity could cause collateral damage through inflammation and edema. For these reasons, we believe care should be taken in choosing the right TAA for which the CAR will be expressed in a gated system.

5. Stylistic comment #1: it would be helpful to the reader if the title addressed the principle conclusions of the study – namely that a SynNotch circuit can reduce ON target, OFF tumor toxicity in a metastatic neuroblastoma model.

We hope that the reviewer would agree with our current title, given the new experiments that have been performed.

6. Stylistic comment #2: Overall the figures are accurately and helpfully labeled and include appropriate error bars, but the figure labels are too small to read across the board.

We have done our best to increase the fonts whenever possible. The editorial staff will also likely increase the font size to their standards for publication.

Reviewer #2:

The manuscript by Moghimi et al. presents a novel application of the recently introduced SynNotch module for tailoring anti-tumor CAR-T cell responses. While CAR-T cells (CARTs) targeting CD19 are in clinical use for B cell derived malignancies, their long-term efficiency against solid tumors is limited due to exhaustion of the transferred T cells and subsequent lack of persistence or by side effects including cytokine-release syndromes and neuropathies (referred to as ICANS, immune cell-associated neurologic syndrome). These limitations might be overcome by the recently introduced synthetic notch module (SynNotch) allowing for coupling of two different tumor-associated antigens (TAAs) to increase selectivity of the CAR-T response while possibly reducing side effects. The authors used this approach to link the TAAs GD2 and B7-H3 via SynNotch. The resulting CARTs were analyzed in the context of neuroblastoma, a solid tumor of childhood, which usually expressed both GD2 and B7-H3.

General remarks:

Work in this manuscript appears to be performed technically sound. Results are presented in a way that enables other researchers to reproduce the work.

That said, I had a hard time following the logic in the current version of the manuscript, which first tells the reader that B7-H3 directed CARTs cure metastatic neuroblastoma, which is in line with results published earlier this year on other solid tumors including childhood malignancies. Second, we learn that re-engineering of CARTs targeting GD2 did not work and only then we are informed on the results of linking GD2 and B7-H3 via SynNotch. I consider the following order as easier to follow: 1. Engineering GD2-CAR T cells by introducing CD28 or 4-1BB co-stimulatory molecules does not work due to toxicity issues and lack of efficacy, respectively; 2. As GD2-CAR T cells have been proven efficacious on NB models and given the encouraging results of others using B7-H3 CARTs, the authors compared the GD2-SynNotch-B7-H3 CAR-T cells to CAR-T cells expressing B7-H3 only.

We thank the reviewer for the suggestion and have now generated additional experiments for the neurotoxicity models with GD2 murine CARs and present this data in Figure 1 prior to addressing the gated B7-H3 and B7H3 work.

Major points:

1) GD2-SynNotch-B7-H3 CAR-T cells yield favorable biomarker and metabolic profiles, while delaying tumor clearance compared to CARTs targeting B7-H3 only. If the authors want to argue for their SynNotch construct on the ground of a favorable molecular profile, they need to add data showing a benefit over the "B7-H3 only" constructs, e.g. by re-challenging the mice with tumor cells upon tumor clearance after 4 weeks. Emergence of exhaustion markers in the "B7-H3 only" CARTs could otherwise be interpreted as a sign of lack of target cells. Therefore, a re-challenge could inform on long-term tumor control conferred by either of the CARTs used here. Additionally, it would also be interesting, if the assumed exhausted phenotype of B7-H3 only CAR-T cells would be reversed by simultaneous treatment with antibodies directed against the identified marker proteins, e.g. LAG-3 and PD-1.

We appreciate the insight of the reviewer on this important issue. To address the benefit of metabolic fitness of the GD2-SynNotch-B7H3 CAR-T cells over B7H3 CAR, we performed an experiment where gated and conventional CAR-T cells were co-cultured with target NBL cells for an extended time course. We frequently replenished target cells to maximize CAR-T activation and push the cells to exhaustion. At the end of the time course, we show that gated T cells express lower levels of classic exhaustion markers. We then adoptively transferred these "exhausted" T cells to our metastatic CHLA255 NBL mouse model. We detected significant T cell expansion in animals treated with the gated CAR-T cells,

and three out of four mice were cured of their tumor in contrast to conventional CAR-T cells that had a minimal expansion, and all mice showed tumor growth. This novel finding suggests that gated CAR-T cells likely have an advantage over conventional CARs especially in solid tumors due to their favorable metabolic and exhaustion profiles. These data are shown in Figure 5F, G, H, and page 19 of the result section, and discussed on page 28, the second paragraph.

2) Higher risk for development of ICANS have been suggested to be linked to CD28-containing constructs, which has been reported in at least two major studies (Neelapu et al., NEJM, 2017 and Koechenderfer et al, JCO, 2015). Additionally, the observation that rapid expansion of CARTs containing CD28 has been linked to breakdown of the blood-brain-barrier leading to fatal outcomes in the JCAR015 trial could have come as a warning sign. The authors are strongly encouraged to rephrase the entire paragraph on p9 accordingly and also to critically discuss these findings. Nevertheless, I consider the results presented in figure S3 as highly important as they demonstrate a link between (rapid expansion of) CD28-containing CARs and ICANS in a solid tumor model. Importantly, the authors need to describe the immune cell infiltrates by additional staining for macrophages and activated microglia (e.g. CD68). It would also be interesting to know, if the CD3-positive cells found in the brain were CAR-T cells.

We appreciate the reviewer's comments and references provide. In the revised manuscript, we expand on the generation of the novel murine CAR-T cells using two different costimulatory domains (CD28 and 4-1BB) and show evidence of GD2 CAR-T mediated neurotoxicity in immunocompetent and immunodeficient murine NBL model, including direct evidence of CAR-T cells presence in mice brain. We also now present data using flow cytometry that, indeed the T cells observed in our immunohistochemical analyses are all CAR T-cells in both of our immunocompetent and immunodeficient models (Figure 1 and Supplemental Figure 2 and 3). We have also expounded our discussion about our findings as it relates to those described by Neelapu et al., NEJM, 2017 and Koechenderfer et al., JCO, 2015. (discussion first paragraph page 25)

3) The authors stress that targets B7-H3/CD276 and GD2 are expressed independently of the MYCN oncogene. As MYC proteins confer an immunosuppressive environment, did the authors observe any differences in cytokine profiles in tumors with high levels of MYCN vs low levels of MYCN? In the metastatic model presented here, intravenous injection of tumor cells was used, which does not fully reflect the microenvironment of a growing tumor. It would be interesting if MYC-related differences were to be observed in s.c. transplant model or, if available, in one of the available MYC-transgenics for neuroblastoma.

We agree with the reviewer that understanding differences in response to low and high MYCN tumors would be interesting. However, this would require several models for each group to be established and evaluated in NSG models. Our metastatic models do form tumors in the liver, but the reviewer is correct that better models (either s.c. or orthotopic injections) may be more useful. This work would also be limited in a purely immunodeficient model, as there are limited murine immunocompetent models for MYCN amplified and non-amplified available for study. We believe these goals are ideal for a future project.

4) Fig. S6: This is an important experiment regarding leakiness of the B7H3 CAR. Unfortunately, data interpretation is hampered by incomplete labelling (no description of the x-and y-axis in A) and description of the experimental read-out. Please give scores of Phox2B and CD3 positive cells to evaluate

if there is a significant difference in tumor cell number and T cell infiltration depending on E:T ratios of UT and the CD123/B7H3 CAR.

We apologize for the lack of labeling and this has been corrected. There were no observable T cells in the tissues obtained in that experiment. Our new *in vivo* experiments utilizing LAN6 with and without forced GD2 expression provides additional information about the lack of clinical significance in a leaky expression of CAR for the GD2⁺B7H3 CAR-T cells (Figure 4E and F). We tested the GD2⁺B7H3 gated CARs in the metastatic NBL model established from the newly created luciferase labeled LAN6 cell lines. LAN6 does not have any GD2 expression, and we forced expressed GD2 thru transduction of GD2 and GD3 synthase gene in LAN6 cell lines - see the response to reviewer #1 Comment #2). We did not observe any clinical benefit in our new *in vivo* system utilizing GD2 negative LAN6 model with the infusion of GD2⁺B7H3 gated CARs. In contrast, we show a significant response in our mice model using the LAN6^{GD2+} cell line (Figure 4F). We speculate that leakiness due to promoter free expression of the CAR protein or due to syn-notch activation cells does not contribute to a sufficient number of T cells that can mount an effective anti-tumor response.

Minor:

several typos need to be addressed, e.g. p 33 (Biolenged = Biolegend), legend to Fig. S6 (in = is).

These have now been corrected.

Reviewer #3:

The authors describe the pre-clinical development of a CAR T-cell therapy approach for childhood solid tumour neuroblastoma. As surface antigens that have a truly tumour specific expression pattern and hence provide optimal targets for a CART approach are limited, they focus on developing a CAR incorporating a SynNotch gate - which is effectively an 'AND' gate - limiting CAR T-cell activation to the presence of two tumour associated antigens in this case GD2 and B7-H3. First, they demonstrate efficacy - in the absence of toxicity - of B7-H3 directed CAR T-cells in an NSG model of metastatic neuroblastoma. Then they describe, in a orthotopic syngeneic neuroblastoma model, that mice treated with CAR T-cells directed to GD2 using a 14g2a-based CAR develop toxicity which they interpret as on-target off-tumour toxicity. They then use a SynNotch approach, referred to as GD2[^]B7-H3 CAR, to create an AND gate with expression of the B7-H3 CAR gated by binding to an affinity enhanced version of the GD2 scFv. Efficacy of GD2[^]B7H3 CAR T-cells is demonstrated in a metastatic model of neuroblastoma although tumour control is reduced as compared to standard B7-H3 CAR T-cells. GD2[^]B7-H3 CAR T-cells were shown to have a reduced oxygen consumption rate and reduced expression of immune inhibitory receptors/transcription factors associated with T-cell exhaustion as compared to B7-H3 CART cells. This metabolic/phenotypic profile of GD2[^]B7H3 may translate into improved in vivo CAR T-cell persistence.

Major comments:

The crux of this paper is the development of an AND gate to minimize risk of on-target off-tumour toxicity of CAR T-cells directed against an tumour associated antigen which is also expressed on some normal tissues. The SynNotch approach to create an AND gate is not novel and has been described by Srivastava et al, Cancer Cell 2019 [PMID:30889382] to prevent on-target off-tumour toxicity albeit in the context of a different tumour type/tumour associated antigens. The authors start out with describing CARs targeting two single antigens expressed on neuroblastoma B7-H3 and GD2. They describe the efficacy of a B7-H3 specific CAR in NSG models of neuroblastoma. Again this is not novel as Du et al [Cancer Cell 2019, PMID: 30753824] have described efficacy of B7-H3 CAR T-cells in the same CHLA255 model of neuroblastoma albeit using a CAR based on a different B7-H3 specific antibody. Similar to described findings by Du et al the authors do not see any treatment related toxicity in this model. The authors then report an unexpected finding using a GD2-directed CAR based on GD2-specific antibody 14g2a. Here a different model of neuroblastoma is used i.e. a syngenic model with renal capsule engraftment of a murine neuroblastoma cell line transduced to express GD2 were mice are treated with chemotherapy prior to CAR T-cell administration. They report that mice treated with T-cells transduced with a GD2-CAR incorporating CD28zeta endodomains developed signs of neurotoxicity whereas toxicity was not observed in mice treated with T-cell transduced with GD2-CAR incorporating 41BBzeta endodomains. The authors interpret signs of toxicity as on-target off-tumour toxicity however, the cause of the observed toxicity is not sufficiently explored to draw this conclusion. Signs of neurotoxicity can be caused through different mechanisms including (1) neurotoxicity secondary to CART activation/inflammation as described in patients treated with CD19-specific CAR T-cells, (2) local inflammation due to homing of CAR T-cell to tumour deposits in the brain or (3) on-target off-tumour toxicity due to GD2 expression on normal brain tissues. The data shown to explore the cause of the observed toxicity is limited to immunohistochemistry of cerebellum/spine of one mouse in the GD2-CAR_CD28zeta endodomain group (Supplementary Figure 3). This showed presence of CD3+ T-cells - however no comparison of the same staining for mice from the other treatment groups is provided. No specific staining for CAR expressing T-cells is performed. It is not established if T-cells are infiltrating in GD2-expressing areas of the CNS. A second query about the result of this experiment is that the tumour burden at baseline does not appear equal between treatment groups. A graph showing quantified BLI signal over time should be included in addition to individual BLI images. The authors then go on to

generate a syn-Notch AND gate with B7-H3. They do not demonstrate on-target off-tumour toxicity with B7-H3 and Du et al (their ref 26) shows no on-target off-tumour toxicity with B7-H3 in an immunocompetent mouse model. Given clinical experience with (plain) 14g2a and no published data showing on-target off-tumor toxicity, there is relatively little justification of making a GD2/B7-H3, especially since the potency of the gated CAR is lower.

The model described in 2016 by Windal Lim's lab has allowed multi-targeting of tumors and their paper focused on leukemias. We believe much remains to be understood about this model in other malignancies, especially solid tumors in children. The gated ROR1 CAR (Srivastava et al 2019 Cancer cell) demonstrated rescue from toxicity from conventional ROR CAR-T cells but showed very little efficacy data in a breast cancer model. Our paper showed the efficacy of the gated CAR-T cells for the first time against a very aggressive neuroblastoma model, although with lower cytotoxic efficiency leading to lower potency than a non-gated version. We also demonstrated novel findings of the superior metabolic and immunologic fitness in this gated T cells in comparison to conventional counterparts, and provide additional *in vivo* data in this regard (see comment to Reviewer 2 - Comment 2)

We now present additional neurotoxicity data for our GD2 CAR T cell, including flow cytometry data demonstrating the trafficking of CAR T-cells into the brain of animals along with proper controls (Figure 1). The ScFv used by Due et al. showed that mB7H3 CAR was only cytotoxic against mB7H3 if expressed in human cells but not against murine cells expressing endogenous B7H3. Thus, it is difficult to extrapolate safety data from murine CAR-T cells with suboptimal cytotoxicity against murine B7H3+ cells (endogenously expressing cells). B7H3 CAR-T cells safety in humans is unknown, but currently these trials are ongoing. There are extensive data demonstrating toxicity with anti-GD2 antibody (pain and neuropathy in the peripheral nervous system). GD2-CAR trial did not suggest any toxicity but also failed to show any efficacy; similarly, we also see a lack of effectiveness and no toxicity with our murine anti-GD2 BBZ CAR. It is unknown if improved anti-GD2 human cars would cause neurotoxicity. However, we believe this to be an important concern given publications demonstrating neurotoxicity with high-affinity anti-GD2 CAR T cells and our data showing toxicity with standard anti-GD2 CAR T cells.

In summary, the main questions are:

1) Why was a different model used to assess the GD2-CAR?

We had initially started to develop murine immunocompetent models to study their effectiveness within immunocompetent and immunodeficient animals and assess the tumor microenvironment. Our experience with neurotoxicity in GD2-CD28 CAR T-cell led us to develop the gated system. The B7H3 we utilized in our gated model does not cross-react with mB7h3. As noted above, the murine anti-B7H3 used by Due et al also does not cause cytotoxicity with murine cells expressing endogenous mB7H3. Hence, we chose to study the gated system in the xenograft model and utilize human CAR-T cells as this also has translational usefulness. These additional data are now part of the main paper (Figure 1 and Supplemental Figures 2 and 3)

2) How can the authors demonstrate that the toxicity seen in this model is due to on-target off-tumor toxicity.

We now present data using flow cytometry and IHC from an additional *in vivo* experiment demonstrating CAR T-cell presence in brain tissue. GD2 is a well-known antigen in the brain, thus presence of our CAR T-cells around neurons strongly suggests direct cytotoxicity. As this phenomenon

also occurs in immunodeficient mice, the likely primary source of toxicity remains CAR T-cells. It is unlikely that cytokine storms are the main culprit in this neurotoxicity, as we do not observe T cell trafficking into the brain nor any evidence of neurotoxicity in mice treated with human or mouse untransduced T cells nor by B7H3 CARs. GD2 CAR neurotoxicity correlates with efficiency in clearing tumors regardless of the models used, suggesting target specific expansion and subsequent toxicity. Additional data generated showed that animals treated with reverse gating strategy B7H3⁺GD2 T cells did not exhibit significant neurotoxicity despite T cell trafficking in the brain, likely due to decay in CAR expression (Figure 6 and discussion Page 26). We agree that role of microglial activation and cytokine production are key areas that require further research and ideal for a future project.

3) As there is existing clinical data (Louis et al, Blood 2011 [PMID:21984804] targeting GD2 in neuroblastoma where neurotoxicity was reported, how relevant as the described findings in this mouse model?

Phase I clinical trial in NBL using anti-GD2 CAR-T with OX-40 costimulation signaling did not show neurotoxicity nor efficacy. Our preclinical findings and published work (Richman et al. 2018 Cancer Immunology Research) suggest there is a correlation between efficiency and toxicity. New clinical trials using GD2-BBz CAR-T cells are underway with a built-in safety mechanism, but these trials have not opened yet. It remains to be seen if neurotoxicity will occur in humans with a high efficacy GD2 CAR T-cells.

4) The generation of a B7-H3/GD2 AND gate CAR is not justified, this would be a logical approach if expressing non-cross reactive on-target toxicity between the two target antigens - otherwise restricting B7-H3 recognition to GD2 expressing tissue risk increasing antigen escape.

Nothing is known about the mechanism of antigen escape within the gated CAR T-cell systems. As these cells are only activated (B7H3 CAR expression) with the GD2 gate, they are not continually circulating and pressuring tumors to escape. Having a gated CAR does reduce the on-target off-tumor toxicity and we demonstrate this now with new data using the reversal of the gating strategy with B7H3 as the gate and GD2^{E101K} as the CAR.

Minor comments:

Figure 1b: left - how are CAR T-cells detected, right; please include control of CAR T-cells co-cultured with a non-B7-H3 expressing cell line

CAR-T cells are detected using an anti-Fab antibody. This is now clarified in the Methods flow cytometry. We have now included co-culture with the Raji cell line that does not express B7H3.

Figure 1c-f: while cytokine secretion, T-cell proliferation and cytotoxicity of B7-H3 CART cells is demonstrated as compared to non-transduced T-cells, no non-B7-H3 expressing target cells are included to demonstrate B7-H3 specificity of CART activity.

We show that B7H3 CAR-T cell cytotoxicity is specific to B7H3 expressing cell lines, and in co-culture experiment with Raji cell line that does not express B7H3, no killing was observed (Figure S4C, D).

Figure 2: G-I shown is T-cell proliferation, cytokine secretion, and cytotoxicity of GD2⁺/B7-H3⁺ CHA255, but not absence of CART activity against a GD2⁻/B7-H3⁺ cell lines such as LAN-6.

We now show this data in Figure 4 D, E, F, and Fig S7C, D where we demonstrate the discriminatory power of gated CAR-T cells to differentiate GD2⁺B7H3⁺ from GD2⁻B7H3⁺ NBL.

Figure 3J: the colors use for mice treated with a single or repeated injection of GD2⁺B7-H3 CART is similar and hence this Figure is hard to read.

This is now included in Figure 4H. We have now adjusted the colors for better visualization.

Figure S3: how were murine GD2-specific CARs constructed?

Please refer to the additions to the Method section (page 35 second paragraph) and Figure 1 and Figure S2. In summary murine G2a anti-GD2 antibody scFv was cloned into CD19-28z using fusion cloning, and later CD28 signaling domain was swapped with 4-1BB domain to construct murine GD2-BBz with a murine CD8 hinge.

Discussion: 'use of the SynNotch gated system was not associated with neurotoxicity even though the SynNotch receptor contains the high affinity GD2(E110K scFv) - the SynNotch receptor was tested in the CHLA NSG model whereas toxicity was observed in the syngeneic NB9464D(GD2+luc+) model. Moreover, to test this as outlined above, a B7-H3⁺GD2(E110K) should be used.

We have now generated B7H3⁺GD2^{E101K} (Figure 6) and show B7H3⁺GD2^{E101K} was effective in eliminating NBL with no clinical symptoms of neurotoxicity. Although activated B7H3⁺GD2 T cells do find themselves in circulation and infiltrate mice brain likely as a result of the expression of the GD2^{E101K}-CAR. However, lack of neurotoxicity is hypothesized to be due to the decay in expression of this CAR which starts after disengaging from the gated antigen. Please also response to Reviewer #1, Comment 1). These findings are now discussed in the paper (Page 26 and 27).

Reviewer #4 (Remarks to the Author):

In this study Moghimi B. et al. report that a SynNotch gated CAR-T cell targeting GD2 as the gate and B7H3 as the target of the CAR controls Neuroblastoma tumor growth in a xenograft mouse model of Neuroblastoma. Moreover, in vitro GD2⁺B7H3 CAR-T cells co-cultured with NBL cells are more resistant to exhaustion and show greater metabolic fitness, compared to 'conventional' B7H3 CAR-T cells.

In my opinion, the manuscript is well written, easy to follow and present original and interesting data. The SynNotch approach is an interesting novel approach that will be of interest. It gives extra specificity to CAR-T cells. GD2⁺B7H3 CAR-T cells show in vitro high NBL killing efficacy, although the in vivo therapeutic efficacy shown is not so striking, since 'conventional' B7H3 CAR-T cells can reach the same potency. Also, the therapeutic efficacy of CD19 enriched GD2⁺B7H3 CAR-T is higher when therapy is started earlier (7 days post i.v. challenge with tumor cells), although still not as efficient as 'conventional' B7H3 CAR-T therapy. Efficacy of CD19 enriched GD2⁺B7H3 CAR-T is improved when three doses starting at day 14 after i.v. tumor cell injection are administered, demonstrating tumor growth control, prolonged mice survival, although all mice finally die.

It seems the most effective therapeutic CAR-T in vivo approach is still that with 'conventional' B7H3 CAR-T, in spite of higher expression of exhaustion markers and inferior metabolic fitness compared to GD2⁺B7H3 CAR-T.

No data are presented on how long GD2⁺B7H3 CAR-T cells or any CAR-T cells can survive in mice and thus can be detected in blood. Can CAR-T cells be recovered in vivo?

It would be interesting to see also the Kaplan-Meier curve associated with the bioluminescence images in all figures.

My major concern resides on the issue of toxicity: since the GD2 associated neurotoxicity is detected in the syngeneic model, the final demonstration that SynNotch modification help to avoid toxicity needs to be demonstrated in the same syngeneic immunocompetent NBL model. In ref 26 and in the discussion the authors properly cite the in vivo use of a B7H3 CAR-T approach in an immunocompetent mouse model, so it seems feasible to use also GD2⁺B7H3 CAR-T. In addition, in the syngeneic model also CAR-T associated toxicity can be studied (circulating cytokines).

Moreover, the author state that CD28 costimulatory domain is associated with neurotoxicity (figure S3 and discussion) as mice receiving GD2-28z CAR-T showed deep toxicity, while mice receiving GD2-BB CAR-T did not. How can the CD28 costimulatory domain be responsible for neurotoxicity?

Just a comment: An additional improvement of SynNotch CAR-T cells may be the insertion of a safety switch that can induce CAR-T cell apoptosis to control their in vivo expansion and prevent or limit any adverse reaction, especially in view of a future translation to the clinic.

Specific comments:

1) Have the in vivo experiments been performed more than once?

Yes, the gated CAR-T cells have been tested in 6 separate cohorts of animal experiments with the added experiments now presented in Figures 1 and 4 and Supplemental Figures 8.

2) In the Introduction section, at the bottom of the first page the authors mention Fig. 2A to explain the SynNotch gating strategy. It sounds strange to me to claim in the introduction "Fig. 2", while Fig 1 has not been mentioned yet.

We appreciate the reviewer's astute observation. We have now generated additional experiments for the neurotoxicity models with GD2 murine CARs and present this data as Figure 1 prior to addressing the gated B7-H3 and B7H3 work. The reference noted has been removed.

3) In the Results section, in the paragraph entitled: "Construction of highly specific and effective GD2/B7H3 CAR-T cells": since you show GD2-28z CAR-T neurotoxicity which is the basis for your future choice of the SynNotch approach, this should be a figure, not a supplementary Figure, see point 4; in Figure S3, in panel A description and in Materials and Methods the therapeutic regimen given to mice before administration of GD2-28z, GD2-BBz CAR-T cells or UT is described. Why did the authors decide to treat syngeneic mice with cyclophosphamide and topotecan before CAR-T administration? No mention or explanation in the Result section about the chemotherapeutic regimen is given, and it should.

We thank the reviewer for the comment and have revised the manuscript and brought forth the neurotoxicity data as Figure 1 and added additional data about our murine GD2 CARs. We also include a brief explanation of our use of chemotherapy in our syngeneic model in methods and results. Briefly, as these mice have a competent immune system, we wanted to simulate the actual treatment given in humans with CAR T-cells, which nearly always includes lymphodepletion with chemotherapy prior to CAR T-cell infusion. Lymphodepletion is crucial to achieving T cell expansion and clinical benefits in patients (Knochelmann, et al, Frontiers in immunology 2018). We have now added a sentence in the Methods section as well as Results stating our intent and use of chemotherapy.

4) In the discussion section:

- "...herein we demonstrate the novel finding that neurotoxicity" Sounds a bit strange since the Figure that reports neurotoxicity in mice receiving GD2 CAR-T cell therapy is a supplementary figure. This should be a regular figure. In the same sentence you imply that CD28 costimulatory domain in GD2-CAR-T is associated with neurotoxicity.

Have you repeated the in vivo experiment more than once with a different GD2-28z CAR-T preparation?

We have now brought the neurotoxicity data as Figure 1 and added additional data (and repeated the experiment). The same results were obtained in a repeated experiment where we treated the orthoptic immunocompetent model of murine NBL with a freshly made batch of murine GD2-28z and GD2-BBz CAR-T cells with the same findings (Figure 1B). We used this experiment to quantify the presence of T cells and document the detection of CAR-T cells in single-cell dissociated brain tissue harvested from 3 mice that were sacrificed around the same time as other animals in the cohort that were found dead of neurotoxicity. We also observed and report on similar neurotoxicity in the metastatic murine NBL model in immunodeficient mice (NSG) treated with murine GD2-28z CAR-T cell and finding of CAR T cells in the brain tissue (Figure 1D).

5) It is not clear how you demonstrate that CD28 costimulatory domain may be responsible for neurotoxicity. If GD2 and CD28 associated toxicity is the reason claimed in the manuscript to be reduced/eliminated, you might not need to use a SynNotch gated strategy, but only modify (as you did) the costimulatory domain from CD28 to 4.1BB. In fact, the authors report that mice treated with GD2-BBz CAR-T cells did not show neurotoxicity, but a limited therapeutic effect. The authors should better clarify this issue.

We have shown previously that neurotoxicity in mice model of human disease is only seen in high-affinity anti-GD2 scFv with BBz costimulation (Richaman et al, Cancer Immunology Research, 2018). The toxicity of a CAR T-cell is likely dependent on multiple factors, including binding-affinity of the scFv, co-stimulatory domain, level of CAR expression, as well as the amount of tumor antigen that allows for CAR T-cell expansion. Additionally, we now discuss the previous report of differences in the neurotoxicity induced by CD19 CAR-T cells thought to be related to the CD28 hinge and transmembrane domain (page 26). Our findings further support the concept that structural features in CAR construction can influence toxicity. Accurate determination of the therapeutic window given the interplay among multiple components of CAR-T cells may be difficult to establish and could vary across individuals. We believe strategies such as SynNotch will have a better safety profile while also aware that improvements need to be made in their efficiency. We now expand on this in our discussion.

5) In the materials and methods section: no indications of how intracellular staining for cytokine production is performed, but in the figures (Fig. 1C, 3E) a contour plot is shown as representative. In general, are data represented in the histogram charts the mean values +/- SD?

We have now added the description for detection of intracellular staining for cytokine production in the Methods section. We used Biologend reagents and followed manufactured protocol (page 39, second paragraph). Data represented in the bar plots reflect mean \pm SD; The cytotoxic incucyte assays (time course experiments) also represent average expression for at least 3 independent experiments with SD. (These are now stated in the legend);

Minor comments:

In the results section:

- second page "...100% overall survival over the 6 month observation period while the mice that received with UT cells or..." Is the word 'with' a misprint?

This has now been corrected. Thank you.

- A line below: "...Similar in vivo efficacy of B7H3 CAR-T cells was observed a second metastatic murine..." Is 'in' missing?

This has now been corrected. Thank you.

- Fig. 2B should be cited when explaining the GD2⁺BFP T cells.

This has now been corrected and cited on page 12.

- In "Enrichment of GD2⁺B7H3 T cells improves efficacy against NBL" can the authors explain what they mean with 'serving as a kill switch' in the sentence "...adding a truncated version of human CD19 (tCD19) to allow for enrichment and detection of T cells while also serving as a kill switch (Fig. 3A)."?

Blinatumomab, a Bi-Specific Anti-CD19/CD3 BiTE, can also recognize the tCD19. Blinatumomab is currently used to treat patients with pre-B aLL. This FDA approved drug could be used to terminate the

gated CAR-T cells expressing truncated CD19 in the unforeseen circumstance of toxicity from the gated CARs. We have now modified this sentence to clarify our meaning.

- The authors demonstrate the specificity of the GD2⁺B7H3 CAR-T SynNotch towards CHLA255 a GD2⁺B7H3⁺ NBL cell line and use as control LAN6 a GD2⁻B7H3⁺ NBL cell line. The control cell line should be shown not only for the cytotoxicity assay, but also for cytokine production.

Figure S7C, D shows intracellular cytokine expression levels in GD2⁺B7H3 CAR-T cells co-cultures with CHLA255 and LAN6.

REVIEWER COMMENTS

Reviewer #1 (Remarks to the Author):

The authors successfully constructed a B7H3[^]GD2 AND gate and compared it to a GD2[^]B7H3 AND gate and they also engineered a LAN6 GD2 KI line for a true measure of the specificity of their system. Based on these substantial revisions I recommend the manuscript for publication. The ability to reengineer the system using the B7H3[^]GD2 gate is impressive and demonstrates the versatility of their chosen system. Overall this is a very promising and impressive study.

The in vivo data using LAN6 vs. LAN6/GD2+ cell lines are particularly revealing. I am curious why E:T ratios of 5:1 and 10:1 were chosen – are similar results observed with E:T ratios of 1:1 or 2:1 – but this is merely curiosity and not a suggestion for additional data.

One final note: the manuscript would be substantially improved if the size of the labels in all figures were increased for legibility.

Reviewer #2 (Remarks to the Author):

The authors have successfully performed the additional experiments that I proposed and they have realised the suggested changes.

Reviewer #3 (Remarks to the Author):

While the manuscript provides data showing that GD2[^]B7-H3 SynNotch CAR T-cells targeting can result in activity dependent on presence of both tumor antigens as well as efficacy - albeit at a reduced level compared to plain B7-H3 CAR - crucially the rationale for using this approach remains not addressed in the revised manuscript. Using a SynNotch gate adds complexity to the CAR approach and hence this needs to be well justified.

There is no data provided to show that targeting B7-H3 results in toxicity and hence using a GD2-gate is not justified. The authors argue that the possibility of toxicity when targeting B7-H3 has not been sufficiently assessed by Du et al. In my view this is a detailed and comprehensive study which uses a B7-H3 binder that cross-reacts between murine and human B7-H3. They report on extensive studies of expression of B7-H3 in normal tissues - both in human and mouse and found that the limited expression on normal tissues was similar across species. They demonstrate binding of the B7-H3 CAR to normal murine cells with endogenous expression of B7-H3 (activated BM-derived DC) by their B7-H3 CAR. Hence, this model appears an appropriate setting to assess (on-target off-tumor) toxicity of targeting B7-H3 with a CAR approach and toxicity was not found in this model. The authors do not provide new information/data to demonstrate that targeting B7-H3 with a CAR results in toxicity.

The SynNotch approach proposed here is designed to prevent activation of the CAR upon recognition of B7-H3 on normal tissues using a second tumor antigen similarly expressed on neuroblastoma but with non-overlapping expression on normal tissues. In this context the statement in the discussion that 'our choice of GD2 as the gatekeeper for B7H3 CAR expression was based on the presence of GD2 in the brain and the desire to eliminate any possibility of CNS toxicity' does not appear logical or does not provide a rationale for the approach taken.

I note that the data on observed neurotoxicity in mice treated with GD2-CAR is now presented in main Figure 1 and additional data on this murine GD2-CAR is provided in supplementary Figure 2.

For the BLI images as presented in Figure 1, BLI quantification is now provided in Figure S2 panel G. For easy of reading it will be useful to include this figure panel in main figure 1. For clarification, please mark on this BLI quantification spider plot at which time point the CART cells were administered. In addition, please include BLI quantification for the other figures showing BLI images as well in the same way (i.e. Figure 2G, 3J, 5J and 6D).

In addition, for the top panel of figure 1B please label which treatment groups the different dot plot figures represent.

Further comments on the rebuttal provided of the main issues raised:

1) Why was a different model used to assess the GD2-CAR?

We had initially started to develop murine immunocompetent models to study their effectiveness within immunocompetent and immunodeficient animals and assess the tumor microenvironment. Our experience with neurotoxicity in GD2-CD28 CAR T-cell led us to develop the gated system. The B7H3 we

utilized in our gated model does not cross-react with mB7h3. As noted above, the murine anti-B7H3 used by Due et al also does not cause cytotoxicity with murine cells expressing endogenous mB7H3. Hence, we chose to study the gated system in the xenograft model and utilize human CAR-T cells as this

also has translational usefulness. These additional data are now part of the main paper (Figure 1 and Supplemental Figures 2 and 3)

> to assess if observed on-target off-tumor toxicity can be prevented with a SynNotch approach the plain GD2-CAR needs to be compared with a the B7-H3 gated GD2-CAR in a model where evidence of toxicity is observed with the plain GD2-CAR. The data as presented in Figure 6 does not include a group of mice treated with plain GD2-CAR and it does not allow to assessment of the performance of the B7-H3[^]GD2-CAR in preventing toxicity. Moreover, I understand from Suppl Fig 1 that in this experiment 4-1BB is used as endodomain for the CAR whereas findings presented in Figure 1 suggest that neurotoxicity occurs when CD28 is used as endodomain for the GD2-CAR.

2) How can the authors demonstrate that the toxicity seen in this model is due to on-target off-tumor toxicity.

We now present data using flow cytometry and IHC from an additional in vivo experiment demonstrating CAR T-cell presence in brain tissue. GD2 is a well-known antigen in the brain, thus presence of our CAR T-cells around neurons strongly suggests direct cytotoxicity. As this phenomenon also occurs in immunodeficient mice, the likely primary source of toxicity remains CAR T-cells. It is unlikely that cytokine storms are the main culprit in this neurotoxicity, as we do not observe T cell trafficking into the brain nor any evidence of neurotoxicity in mice treated with human or mouse untransduced T cells nor by B7H3 CARs. GD2 CAR neurotoxicity correlates with efficiency in clearing tumors regardless of the models used, suggesting target specific expansion and subsequent toxicity. Additional data generated showed that animals treated with reverse gating strategy B7H3[^]GD2 T cells did not exhibit significant neurotoxicity despite T cell trafficking in the brain, likely due to decay in CAR expression (Figure 6 and discussion Page 26). We agree that role of microglial activation and cytokine production are key areas that require further research and ideal for a future project.

> The provided new data indicate that an increased number of T-cells and CAR T-cells are detected in the brains of the mice which showed symptoms of toxicity. I appreciate that it is challenging to demonstrate if increased CAR T-cell presence in the brain is due to off-tumor antigen recognition and is the cause of the toxicity observed.

Published data on toxicity associated with targeting GD2 has shown conflicting results. In the model described by the authors as well as in work published by Rickmann et al where an affinity enhanced version of the GD2-CAR was used, neurotoxicity has been described. While the observed toxicity may

be caused by on-target off-tumor toxicity it can also be caused by other mechanisms i.e. cytokine-driven (described by Majzner et al [PMID: 29610423] on Rickmann et al). Moreover, other publications that utilized the same high affinity GD2-CAR did not find toxicity (Lynn et al [PMID: 31802004]) neither did other studies in murine models using non-affinity enhanced 14g2a GD2-CAR T cells (Long et al, [PMID: 25939063] and [PMID: 27549124]). I recommend including these published findings in interpretation and discussion of your data shown in Figure 1 of the revised manuscript.

3) As there is existing clinical data (Louis et al, Blood 2011 [PMID:21984804] targeting GD2 in neuroblastoma where neurotoxicity was reported, how relevant as the described findings in this mouse model?

Phase I clinical trial in NBL using anti-GD2 CAR-T with OX-40 costimulation signaling did not show neurotoxicity nor efficacy. Our preclinical findings and published work (Richman et al. 2018 Cancer Immunology Research) suggest there is a correlation between efficiency and toxicity. New clinical trials

using GD2-BBz CAR-T cells are underway with a built-in safety mechanism, but these trials have not opened yet. It remains to be seen if neurotoxicity will occur in humans with a high efficacy GD2 CAR T-cells.

> I agree that further data from ongoing clinical trials are required to determine if GD2 (and B7-H3) can be safely targeted with a CAR approach. Early findings from these clinical trials are appropriately discussed and cited in the manuscript.

4) The generation of a B7-H3/GD2 AND gate CAR is not justified, this would be a logical approach if expressing non-cross reactive on-target toxicity between the two target antigens - otherwise restricting B7-H3 recognition to GD2 expressing tissue risk increasing antigen escape.

Nothing is known about the mechanism of antigen escape within the gated CAR T-cell systems. As these cells are only activated (B7H3 CAR expression) with the GD2 gate, they are not continually circulating and pressuring tumors to escape. Having a gated CAR does reduce the on-target off-tumor toxicity and

we demonstrate this now with new data using the reversal of the gating strategy with B7H3 as the gate and GD2 E101K as the CAR.

> A justification for the requirement of a the SynNotch approach when targeting B7-H3 is required. Not only because the SynNotch approach adds complexity but also as it is reasonable to anticipate that making CAR function dependent on expression of two tumor antigens will increase the risk of tumor escape due to antigen loss. Antigen loss in the leukemia setting both for CD19 and CD22 target antigens has now been well described. Hence, a strong rationale for having to mitigate against toxicity is required as the approach used introduces a new vulnerability. This is an important consideration which is not acknowledged/discussed in the manuscript.

> In the context of targeting GD2 in neuroblastoma, a rationale for using the SynNotch approach is provided in Figure 1 of the manuscript. However, as stated above, the data as shown in Figure 6, in my view does not convincingly demonstrate that using a B7-H3 gate, toxicity associated with the GD2-CAR is prevented.

Minor comments have all been addressed except for additional information required about what is now Supl Fig 1:

Figure S3: how were murine GD2-specific CARs constructed?

Please refer to the additions to the Method section (page 35 second paragraph) and Figure 1 and Figure S2. In summary murine G2a anti-GD2 antibody scFv was cloned into CD19-28z using fusion cloning, and later CD28 signaling domain was swapped with 4-1BB domain to construct murine GD2-

BBz with a murine CD8 hinge.

> could you please state scFv from which GD2 (and B7-H3 antibodies) were used to generate the CAR construct. I understand from the above that for the murine CAR this is the 14G2a antibody but this - I think - is not stated in the methods of the manuscript.

Reviewer #4 (Remarks to the Author):

In this study Moghimi B. et al. report that a SynNotch gated CAR-T cell targeting GD2 as the gate and B7H3 as the target of the CAR controls tumor growth in vitro and in a metastatic xenograft mouse model of Neuroblastoma. Moreover, in vitro GD2⁺B7H3 CAR-T cells co-cultured with NBL cells are more resistant to exhaustion and show greater metabolic fitness, compared to 'conventional' B7H3 CAR-T cells. GD2⁺B7H3 CAR-T cells repeatedly restimulated in vitro with NBL cells showed superior anti-tumor activity once injected in vivo.

In my opinion, authors answered the questions raised. The manuscript is well written, easy to follow and present original and interesting data. All data presented are supported by statistical analysis and data presented are representative of multiple experiments.

Now data are presented in a more logical sequence making it easy to follow the results. Data are discussed appropriately and in full. Methods have improved.

I still have minor questions:

- 1) Are the IVIS Bioluminescence analysis presented in Figures normalized? Could the author show the units of measurement of photons (I.e. ph/sec/cm²/sr) in each figure? Did long living IVIS neg mice remained tumor free at the sacrifice?
- 2) Figure 1: panel B: an indication of which experimental group data are presented would help; panel A and B are wrongly described in the legend; panel C: have you evidence tumor cells are not present?
- 3) page 9 line 10: in the brackets you mention figure S4... panel D? !
- 4) Page 10 lines 8-9 "To evaluate.....B7H3 CAR-T cells." is not fully clear...it is better explained in the legend to Figure 2 panel H.
- 5) in figure S3 "...showing mouse in various regions..." mouse? Are data presented derived from immunocompetent or immunodeficient mice?
- 6) page 13 line 20: "...GD2⁺B7H3 CAR-T cells control showed...." control? Wouldn't it be better to cite Fig. S6 after "tumor bed" (line 20), instead at the end of the sentence (line 21-22)? That figure does not show the comparison between B7H3 CAR-T and GD2⁺B7H3 CAR-T cells.

Finally, given the complexity of the in vivo experiments described in figures the paper would benefit if each figure showing in vivo experiments were accompanied with a scheme indicating tumor cell injection, time of treatments (day+1...) and CART cell used.

Reviewer #3:

Reviewer #3 (Remarks to the Author):

While the manuscript provides data showing that GD2⁺B7-H3 SynNotch CAR T-cells targeting can results in activity dependent on presence of both tumor antigens as well as efficacy - albeit at a reduced level compared to plain B7-H3 CAR - crucially the rationale for using this approach remains not addressed in the revised manuscript. Using a SynNotch gate adds complexity to the CAR approach and hence this needs to be well justified.

There is no data provided to show that targeting B7-H3 results in toxicity and hence using a GD2-gate is not justified. The authors argue that the possibility of toxicity when targeting B7-H3 has not been sufficiently assessed by Du et al. In my view this is a detailed and comprehensive study which uses a B7-H3 binder that cross-reacts between murine and human B7-H3. They report on extensive studies of expression of B7-H3 in normal tissues - both in human and mouse and found that the limited expression on normal tissues was similar across species. They demonstrate binding of the B7-H3 CAR to normal murine cells with endogenous expression of B7-H3 (activated BM-derived DC) by their B7-H3 CAR. Hence, this model appears an appropriate setting to assess (on-target off-tumor) toxicity of targeting B7-H3 with a CAR approach and toxicity was not found in this model. The authors do not provide new information/data to demonstrate that targeting B7-H3 with a CAR results in toxicity.

The SynNotch approach proposed here is designed to prevent activation of the CAR upon recognition of B7-H3 on normal tissues using a second tumor antigen similarly expressed on neuroblastoma but with non-overlapping expression on normal tissues. In this context the statement in the discussion that 'our choice of GD2 as the gatekeeper for B7H3 CAR expression was based on the presence of GD2 in the brain and the desire to eliminate any possibility of CNS toxicity' does not appear logical or does not provide a rationale for the approach taken.

We believe that Du et al. paper is an excellent manuscript. However, the B7-H3 CAR developed in their syngeneic model does not address possible toxicities. In our personal communications with Dr. Soldano Ferrone, who developed the 376.96 antibody utilized as murine scFV in this paper, he confirmed lack of binding to endogenous murine B7-H3. This is the reason a different murine anti-B7H3 antibody was used in Du et al. paper to characterize murine B7-H3 expression by IHC in various tissues. The data on activated BM-derived DC only demonstrated IL2 secretion, no data was presented in terms of cytotoxic effect of this CAR-T cells on endogenous B7H3 expressing cells. The trials of B7H3 CAR-T cells may or may not generate toxicity in humans. Indeed, as we argue in the discussion different CAR-T cell constructs may behave differently in clinic setting when targeting the same antigen. The efficacy and toxicity of any CAR T-cells is dependent on design of the CAR. We reference differential toxicity seen with clinical trial using different CD19 CARs based on the choice of the hinge and transmembrane region of CD28 co-stimulatory domain (Discussion, Page 27). The inducible systems such as SynNotch provide additional safety features. We further show novel findings about improved metabolic fitness and ability to withstand exhaustion after repeated co-culture with tumor cells. Please see revised discussion in Page 25, 26 and 27.

I note that the data on observed neurotoxicity in mice treated with GD2-CAR is now presented in main Figure 1 and additional data on this murine GD2-CAR is provided in supplementary Figure 2.

For the BLI images as presented in Figure 1, BLI quantification is now provided in Figure S2 panel G. For easy of reading it will be useful to include this figure panel in main figure 1.

We have now moved the BLI data into Figure 1.

For clarification, please mark on this BLI quantification spider plot at which time point the CART cells were administered. In addition, please include BLI quantification for the other figures showing BLI images as well in the same way (i.e. Figure 2G, 3J, 5J and 6D).

We have now included all the BLI images raw data and figures within a supplemental data figure as to reduce cluttering of figures in the paper.

In addition, for the top panel of figure 1B please label which treatment groups the different dot plot figures represent.

We apologize for this oversight. This has been corrected.

Further comments on the rebuttal provided of the main issues raised:

1) Why was a different model used to assess the GD2-CAR?

We had initially started to develop murine immunocompetent models to study their effectiveness within immunocompetent and immunodeficient animals and assess the tumor microenvironment. Our experience with neurotoxicity in GD2-CD28 CAR T-cell led us to develop the gated system. The B7H3 we utilized in our gated model does not cross-react with mB7h3. As noted above, the murine anti-B7H3 used by Due et al also does not cause cytotoxicity with murine cells expressing endogenous mB7H3. Hence, we chose to study the gated system in the xenograft model and utilize human CAR-T cells as this also has translational usefulness. These additional data are now part of the main paper (Figure 1 and Supplemental Figures 2 and 3)

> to assess if observed on-target off-tumor toxicity can be prevented with a SynNotch approach the plain GD2-CAR needs to be compared with a the B7-H3 gated GD2-CAR in a model where evidence of toxicity is observed with the plain GD2-CAR. The data as presented in Figure 6 does not include a group of mice treated with plain GD2-CAR and it does not allow to assessment of the performance of the B7-H3[^]GD2-CAR in preventing toxicity. Moreover, I understand from Suppl Fig 1 that in this experiment 4-1BB is used as endodomain for the CAR whereas findings presented in Figure 1 suggest that neurotoxicity occurs when CD28 is used as endodomain for the GD2-CAR.

Figure 1 data is generated using “murine” GD2 CAR-T cell. The anti-GD2 scFV used is non-mutated and it is the CD28 endodomain that causes neurotoxicity in this immunocompetent model. This is the first publication utilizing murine anti-GD2 CAR-T cells in a syngeneic system where a change only in the co-stimulatory domain is responsible for the observed neurotoxicity. As we argue in the discussion, differences in the various structures of CAR-T cells can lead to different phenotypes in terms of efficacy and toxicity. The remainder of the paper utilizes “human” T-cells in constructing CARs, and it is likely that the designs of human and murine CAR-T cells can lead to different phenotypes. All of the human anti-GD2 scFV utilized in human CAR T-cells have mutated E101K anti-GD2 scFV. The E101K anti-GD2 with same design as ours (Dr. Moghimi was co-author on Richman et al.’s paper) caused fatal neurotoxicity in NSG models (Richman et al). A new publication from Richman et al. shows again

neurotoxicity caused by E101K anti-GD2 CAR-T cells with 4-1BB co-stimulatory domain (Richman et al. Molecular Therapy 2020).

The fact that we demonstrate neurotoxicity phenomena in an immunocompetent model using non-mutated GD2 further substantiates the fact that certain GD2 CAR constructs can cause neurotoxicity. The goal of our SynNotch approach in Figure 6 using E101K anti-GD2 with 4-1BB co-stimulatory domain was to demonstrate that the system works with a gated scFV and co-stimulatory domain that has been shown in two publications to cause fatal neurotoxicity within one week. Our data also show that only B7H3-gated GD2 CAR cells, but not other CAR-T cells (B7H3 CARs, GD2-gated B7H3 CARs, or untransduced T-cells) migrate to the CNS. However, as we discussed, the SynNotch system did not induce neurotoxicity due to decay of anti-GD2 scFV in these gated CARs.

2) How can the authors demonstrate that the toxicity seen in this model is due to on-target off-tumor toxicity.

We now present data using flow cytometry and IHC from an additional in vivo experiment demonstrating CAR T-cell presence in brain tissue. GD2 is a well-known antigen in the brain, thus presence of our CAR T-cells around neurons strongly suggests direct cytotoxicity. As this phenomenon also occurs in immunodeficient mice, the likely primary source of toxicity remains CAR T-cells. It is unlikely that cytokine storms are the main culprit in this neurotoxicity, as we do not observe T cell trafficking into the brain nor any evidence of neurotoxicity in mice treated with human or mouse untransduced T cells nor by B7H3 CARs. GD2 CAR neurotoxicity correlates with efficiency in clearing tumors regardless of the models used, suggesting target specific expansion and subsequent toxicity.

Additional data generated showed that animals treated with reverse gating strategy B7H3⁺GD2 T cells did not exhibit significant neurotoxicity despite T cell trafficking in the brain, likely due to decay in CAR expression (Figure 6 and discussion Page 26). We agree that role of microglial activation and cytokine production are key areas that require further research and ideal for a future project.

> The provided new data indicate that an increased number of T-cells and CAR T-cells are detected in the brains of the mice which showed symptoms of toxicity. I appreciate that it is challenging to demonstrate if increased CAR T-cell presence in the brain is due to off-tumor antigen recognition and is the cause of the toxicity observed.

Published data on toxicity associated with targeting GD2 has shown conflicting results. In the model described by the authors as well as in work published by Rickmann et al where an affinity enhanced version of the GD2-CAR was used, neurotoxicity has been described. While the observed toxicity may be caused by on-target off-tumor toxicity it can also be caused by other mechanisms i.e. cytokine-driven (described by Majzner et al [PMID: 29610423] on Rickmann et al). Moreover, other publications that utilized the same high affinity GD2-CAR did not find toxicity (Lynn et al [PMID: 31802004]) neither did other studies in murine models using non-affinity enhanced 14g2a GD2-CAR T cells (Long et al, [PMID: 25939063] and [PMID: 27549124]. I recommend including these published findings in interpretation and discussion of your data shown in Figure 1 of the revised manuscript.

We appreciate the additional references described by the reviewer. As our figure 1 suggests, along with published data referenced by the reviewer, there can exist different GD2 CARs with different efficacy (e.g. due to exhaustion, co-stimulatory domains) or toxicity (e.g. due to E101K as published by Richman et al.), and now our findings in murine model. The E101K GD2-CAR cells used in Lynn et al, has a different structure than Richman et al. Lynn et al paper utilized CD28 co-stimulatory domain and CH2-

CH3 linker which initially showed exhaustion and rescued with c-JUN overexpression; neither caused neurotoxicity. Thi E101K GD2-CAR used by Richman et al incorporated a 4-1BB co-stimulatory domain and no CH2-CH3 linker. Long et al again used a very different 3rd generation co-stimulatory domain (CD28-OX40) in their GD2 CAR. The latest manuscript by Richman et al. in Molecular Therapeutics 2020 (PMID: 32559430) again demonstrated E101K GD2 CAR fatal neurotoxicity.

We have now expanded the discussion (Page 26 and 27) to include the references mentioned. We believe that there may exist a therapeutic window for GD2 CARs, with proper safety mechanism in place, for treatment of GD2+ tumors. Our contention is that assessing toxicities of CAR systems in general may be difficult both due to lack of understanding of the effects of CAR design on downstream signaling and utilization of NSG mice where targets are not cross-reactive to the scFVs used in CAR development. Our findings in murine CAR-T cells emphasize this argument where a simple change in the co-stimulatory domain leads to fatal neurotoxicity. We believe there is room for research and utilization of inducible CAR systems such as SynNotch that offer additional protection against unforeseen toxicities. We also demonstrate additional novel data about this system's superior metabolic fitness and resilience to exhaustion. Additional research should be conducted to improve the efficacy of these inducible systems.

3) As there is existing clinical data (Louis et al, Blood 2011 [PMID:21984804] targeting GD2 in neuroblastoma where neurotoxicity was reported, how relevant as the described findings in this mouse model?

Phase I clinical trial in NBL using anti-GD2 CAR-T with OX-40 costimulation signaling did not show neurotoxicity nor efficacy. Our preclinical findings and published work (Richman et al. 2018 Cancer Immunology Research) suggest there is a correlation between efficiency and toxicity. New clinical trials using GD2-BBz CAR-T cells are underway with a built-in safety mechanism, but these trials have not opened yet. It remains to be seen if neurotoxicity will occur in humans with a high efficacy GD2 CAR T-cells.

> I agree that further data from ongoing clinical trials are required to determine if GD2 (and B7-H3) can be safely targeted with a CAR approach. Early findings from these clinical trials are appropriately discussed and cited in the manuscript.

We wholeheartedly agree with the reviewer.

4) The generation of a B7-H3/GD2 AND gate CAR is not justified, this would be a logical approach if expressing non-cross reactive on-target toxicity between the two target antigens - otherwise restricting B7-H3 recognition to GD2 expressing tissue risk increasing antigen escape.

Nothing is known about the mechanism of antigen escape within the gated CAR T-cell systems. As these cells are only activated (B7H3 CAR expression) with the GD2 gate, they are not continually circulating and pressuring tumors to escape. Having a gated CAR does reduce the on-target off-tumor toxicity and we demonstrate this now with new data using the reversal of the gating strategy with B7H3 as the gate and GD2 E101K as the CAR.

> A justification for the requirement of a the SynNotch approach when targeting B7-H3 is required. Not only because the SynNotch approach adds complexity but also as it is reasonable to anticipate that

making CAR function dependent on expression of two tumor antigens will increase the risk of tumor escape due to antigen loss. Antigen loss in the leukemia setting both for CD19 and CD22 target antigens has now been well described. Hence, a strong rationale for having to mitigate against toxicity is required as the approach used introduces a new vulnerability. This is an important consideration which is not acknowledged/discussed in the manuscript.

We believe that the additional data about resilience to exhaustion and metabolic fitness of SynNotch CAR T-cells make these inducible systems worthy of further study and is of high novelty. Our paper also does not focus nor presents any data on antigen loss in SynNotch CAR T-cells. It is unknown if the inducible systems would increase rate of antigen loss. This system is quite different than CARs directly targeting two antigens, and we believe discussion in this area without any data is purely conjecture. We have stated in the discussion that further research in SynNotch technology is needed.

> In the context of targeting GD2 in neuroblastoma, a rationale for using the SynNotch approach is provided in Figure 1 of the manuscript. However, as stated above, the data as shown in Figure 6, in my view does not convincingly demonstrate that using a B7-H3 gate, toxicity associated with the GD2-CAR is prevented.

The data in Figure 6 demonstrates there is infiltration of CAR-T cells that occurs only in B7H3^{GD2}^{E101K} CARs but not in GD2^{E101K}^{B7H3} CARs nor untransduced T cells. These data show selective infiltration in the posterior fossa where there is high expression of GD2. Given two previous publications where neurotoxicity was fatal within 7-10 days with a CAR dosing and scFv that is exactly as that used in our B7H3^{GD2} SynNotch construct, we believe the SynNotch was successful in mitigating any clinical evidence of neurotoxicity.

Minor comments have all been addressed except for additional information required about what is now Supl Fig 1:

Figure S3: how were murine GD2-specific CARs constructed?

Please refer to the additions to the Method section (page 35 second paragraph) and Figure 1 and Figure S2. In summary murine G2a anti-GD2 antibody scFv was cloned into CD19-28z using fusion cloning, and later CD28 signaling domain was swapped with 4-1BB domain to construct murine GD2-BBz with a murine CD8 hinge.

> could you please state scFv from which GD2 (and B7-H3 antibodies) were used to generate the CAR construct. I understand from the above that for the murine CAR this is the 14G2a antibody but this - I think - is not stated in the methods of the manuscript.

We used the 14G2a in our murine CAR constructs (data from Figure 1). This is now explicitly stated in Methods (Page 37 last paragraph). We did not generate any anti murine B7-H3 CARs nor are there any experiments utilizing murine B7-H3 CARs. We did generate anti-human B7-H3 CARs (sequence derived from patent of MG27A – MacroGenics Inc.). This is now stated in Methods as well (Page 38, second paragraph).

Reviewer #4 (Remarks to the Author):

In this study Moghimi B. et al. report that a SynNotch gated CAR-T cell targeting GD2 as the gate and

B7H3 as the target of the CAR controls tumor growth in vitro and in a metastatic xenograft mouse model of Neuroblastoma. Moreover, in vitro GD2⁺B7H3 CAR-T cells co-cultured with NBL cells are more resistant to exhaustion and show greater metabolic fitness, compared to 'conventional' B7H3 CAR-T cells. GD2⁺B7H3 CAR-T cells repeatedly restimulated in vitro with NBL cells showed superior anti-tumor activity once injected in vivo.

In my opinion, authors answered the questions raised. The manuscript is well written, easy to follow and present original and interesting data. All data presented are supported by statistical analysis and data presented are representative of multiple experiments.

Now data are presented in a more logical sequence making it easy to follow the results. Data are discussed appropriately and in full. Methods have improved.

I still have minor questions:

1) Are the IVIS Bioluminescence analysis presented in Figures normalized?

IVIS BLI presented shown are not normalized and are presented as photons per second (p/s).

Could the author show the units of measurement of photons (i.e. ph/sec/cm²/sr) in each figure?

All figure axis now shows photons per second (p/s)

Did long living IVIS neg mice remained tumor free at the sacrifice? Yes, we now show raw BLI data and plots of BLI for all animals in this study (Supplemental Data)

2) Figure 1: panel B: an indication of which experimental group data are presented would help; panel A and B are wrongly described in the legend;

We have now corrected the legend and describe how panel B is from an additional experiment with groups of mice treated similar to the groups used in survival study (Fig. 1A). The mice from the second set of experiments were sacrificed with the onset of neurological symptoms in GD2-28Z treated group.

panel C: have you evidence tumor cells are not present?

We did not observe any luciferase signal from the brain of the animals in this study but did not stain for tumor cells in the brain. All the groups were injected with same tumor dose at the same time. It would be highly unusual in our repeated experiments to assume that only the animals treated with GD2-28Z CAR-T cells had tumor cells in the brain. It is more likely that the GD2-28z CAR-T cells infiltrated the CNS after *in vivo* expansion.

2) page 9 line 10: in the brackets you mention figure S4... panel D? !

Thank you, this has been corrected.

3) Page 10 lines 8-9 "To evaluate.....B7H3 CAR-T cells." is not fully clear...it is better explained in the legend to Figure 2 panel H.

We have now reworded the description to state: "Immunohistochemical evaluation of liver tissues of mice with high-burden disease (day 28 post tumor inoculation) sacrificed 7 days post B7H3 CAR-T cell infusion revealed impressive T cell infiltration and tumor reduction compared to mice treated with UT

cells (Fig. 2H).”

- 4) in figure S3 “...showing mouse in various regions...” mouse? Are data presented derived from immunocompetent or immunodeficient mice?

These are immunocompetent mice. The title now reads: “**Immunohistochemical analysis of brain tissue from immunocompetent mice treated with murine GD2-28z**”. (Page 31)

- 6) page 13 line 20: “...GD2^{B7H3} CAR-T cells control showed....” control?

Wouldn't it be better to cite Fig. S6 after “tumor bed” (line 20), instead at the end of the sentence (line 21-22)? That figure does not show the comparison between B7H3 CAR-T and GD2^{B7H3} CAR-T cells.

The word “control” has been removed and we agree and moved the Figure citation as suggested by the reviewer.

Finally, given the complexity of the *in vivo* experiments described in figures the paper would benefit if each figure showing *in vivo* experiments were accompanied with a scheme indicating tumor cell injection, time of treatments (day+1...) and CART cell used.

We appreciate the reviewer's comment; We have added the injection time to all the BLI data presented either in the figures or in the supplemental data containing the BLI data not shown in figures.

Nearly all of the regimens for the human CAR-T cells used in our *in vivo* experiments were conducted either as treatment with CAR T-cells on Day +7 or Day +14 after tumor inoculation. Exceptions were:

Figure 1. for immunocompetent mice had chemotherapy given prior to murine CAR T-cells

Figure 4i. Repeated injections of CAR -T cells noted on the figure

Figure 5g. Exhausted CAR-T cells are injected on Day +1

All the legends also describe the timing of the CAR-T cell injections.

REVIEWERS' COMMENTS

Reviewer #3 (Remarks to the Author):

Thank you for including an extended discussion on the difficulties in predicting on-target off-tumor toxicity in preclinical models with reference to published data on targeting B7-H3 to provide a rationale for the use of the described SynNotch approach.

Thank you for providing quantification figures of the BLI data presented.

With this all points are addressed except for my request to discuss the limitations of the SynNotch approach. Alongside the advantages of the SynNotch system which are stated and include reducing the risk of on-target off-tumor toxicity and improving metabolic profile, potential drawbacks of the approach need to be discussed also. Antigen loss of both CD19 and CD22 in the leukemia setting has now been well described and is one of the main reasons of treatment failure with CAR T-cell therapy [PMID: 30275569, PMID: 29155426]. When expression of two target antigens is required for CAR function this may increase vulnerability to antigen loss. This is not conjecture and should be included in the discussion. In this discussion your argument that it is unknown if the inducible system would increase rate of antigen loss equally as compared to CARs directly targeting two antigens can be included.

Reviewer #4 (Remarks to the Author):

The authors answered the questions asked and made some changes. I have a comment: I am very surprised that the IVIS BLI presented are not normalized within the groups, in my opinion they should have been.

Reviewer #3 (Remarks to the Author):

Thank you for including an extended discussion on the difficulties in predicting on-target off-tumor toxicity in preclinical models with reference to published data on targeting B7-H3 to provide a rationale for the use of the described SynNotch approach.

Thank you for providing quantification figures of the BLI data presented.

With this all points are addressed except for my request to discuss the limitations of the SynNotch approach. Alongside the advantages of the SynNotch system which are stated and include reducing the risk of on-target off-tumor toxicity and improving metabolic profile, potential drawbacks of the approach need to be discussed also. Antigen loss of both CD19 and CD22 in the leukemia setting has now been well described and is one of the main reasons of treatment failure with CAR T-cell therapy [PMID: 30275569, PMID: 29155426]. When expression of two target antigens is required for CAR function this may increase vulnerability to antigen loss. This is not conjecture and should be included in the discussion. In this discussion your argument that it is unknown if the inducible system would increase rate of antigen loss equally as compared to CARs directly targeting two antigens can be included.

We appreciate reviewer's constructive comments throughout this peer review process which have strengthened the manuscript. We agree that using SynNotch CAR-T cell targeting a combination of two antigens could potentially increase the chance of antigen escape from downregulation of the either antigens. We have discussed this issue on page 19. Specifically, we state:

"A potential consequence of improved efficacy can be the loss or diminishing density of tumor antigen as has been demonstrated in CAR-T cells directed against CD19 or CD20 positive leukemia cells (PMID: 30275569, PMID: 29155426). The dependence of SynNotch CAR activity on expression of two tumor antigens could potentially increase risk of tumor escape due to loss or attenuation of either antigens. "

Reviewer #4 (Remarks to the Author):

The authors answered the questions asked and made some changes. I have a comment: I am very surprised that the IVIS BLI presented are not normalized within the groups, in my opinion they should have been.

We performed live imaging in all experiments according to the procedure described in the methods section. Specifically, mice were imaged for a constant time (3 minutes) following injection of set amount of luciferase substrate in all of the experiments. The reported average photon/second value over 3 minutes was calculated using the Living Image software version 2.5. The mice were randomly assigned to each group based on the luciferase activity level prior to start of treatment which also ensured that mice from a particular cage could receive different treatments and imaging of different treatment groups occurred together which reduced any bias. The procedures and reporting of BLI used in our publication have been utilized by other groups in similar fashion (e.g. PMID: 30655315, PMID: 31802004, PMID: 30889382). We did find one paper that normalizes BLI to the area of subcutaneous tumor and reported BLI/cm², however this is not achievable in the metastatic models used in this paper. A recent manuscript by Inoue et al ("Timing of Imaging after D-Luciferin Injection Affects the Longitudinal Assessment of Tumor Growth Using In Vivo Bioluminescence", International Journal of Biomedical Imaging 2020) also compared measurements taken over several timepoints after luciferase substrate administration to identify the peak time of bioluminescence. While times to achieve peak luminescence differed based on the day after tumor inoculation, they found extremely high correlation (>0.97) of measurement for photons/second over a single time point vs. peak signal.

We are not completely clear about the reviewer's question regarding normalization per group and assessment over time (decrease in percentage from zero timepoint?), but based on the raw data provided, we believe that any normalization would not yield any measurable change in interpretation of the experiments nor their conclusions.